# ResRL: Boosting LLM Reasoning via Negative Sample Projection Residual Reinforcement Learning

**Zihan Lin** [* † 1 2 3]  **Xiaohan Wang** [* 2]  **Jie Cao** [1]  **Jiajun Chai** [2]  **Li Wang** [2]  **Xiaodong Lu** [2]  **Wei Lin** [2]
**Ran He** [1 3]  **Guojun Yin** [2]

## Abstract

Reinforcement Learning with Verifiable Rewards (RLVR) enhances the reasoning capabilities of Large Language Models (LLMs) but usually exhibits limited generation diversity due to the over-incentivization of positive rewards. Although methods like Negative Sample Reinforcement (NSR) mitigate this issue by upweighting penalty on negative samples, they may suppress the semantic distributions shared between positive and negative responses. To boost reasoning ability without losing diversity, this paper proposes negative sample projection Residual Reinforcement Learning (ResRL) that decouples similar semantic distributions between positive and negative responses. We theoretically link Lazy Likelihood Displacement (LLD) to negative-positive head-gradient interference and derive a single-forward proxy that upper-bounds representation alignment to guide conservative advantage reweighting. ResRL then projects negative-token hidden representations onto an SVD-based low-rank positive subspace and uses projection residuals to modulate negative gradients, improving reasoning while preserving diversity and outperforming strong baselines on average across twelve benchmarks spanning Mathematics, Code, Agent Tasks, and Function Calling. Notably, ResRL surpasses NSR on mathematical reasoning by 9.4% in Avg@16 and 7.0% in Pass@128. Code is available at https://github.com/1229095296/ResRL.git.

---

[*]Equal contribution    [†]Work was done during an internship at Meituan.  [1]MAIS&NLPR, Institute of Automation, Chinese Academy of Sciences, Beijing, China [2]Meituan, Beijing, China [3]School of Advanced Interdisciplinary Sciences, University of Chinese Academy of Sciences, Beijing, China. Correspondence to: Xiaohan Wang <wangxiaohan17@meituan.com>, Guojun Yin <yinguojun02@meituan.com>, Ran He <ran.he@ia.ac.cn>.

*Proceedings of the $43^{rd}$ International Conference on Machine Learning*, Seoul, South Korea. PMLR 306, 2026. Copyright 2026 by the author(s).

## 1. Introduction

Reinforcement Learning with Verifiable Rewards (RLVR) has emerged as a prominent post-training paradigm for enhancing the reasoning capabilities of Large Language Models (LLMs) (Shao et al., 2025; Lu et al., 2026). Notably, DeepSeek-R1 has demonstrated that RLVR can yield significant performance improvements in complex scenarios, introducing the widely adopted Group-Relative Policy Optimization (GRPO) (Guo et al., 2025). However, recent studies indicate that while RLVR effectively optimizes targeted metrics and increases the likelihood of generating high-reward responses, it significantly reduce the base model's output diversity, potentially leading to mode collapse during training (Simoni et al., 2025). Concretely, improvements in Pass@1 accuracy may come at the expense of Pass@$k$ performance; this trade-off may hinder exploration and limit generalization on out-of-distribution tasks (Zhu et al., 2025a; Deng et al., 2025c; Zeng et al., 2024).

To enhance generation diversity and improve Pass@k performance of RLVR, Negative Sample Reinforcement (NSR) has offered an alternative view of policy optimization by explicitly differentiating between positive (high-reward) and negative (low-reward) responses (Zhu et al., 2025b). NSR shifts the optimization paradigm from mainly encouraging the generation of positive responses to actively suppressing negative ones. This approach enables RLVR to enhance model performance (Pass@1) while preserving output diversity (Pass@$k$). However, NSR primarily achieves this by upweighting the gradients of negative responses. We posit that indiscriminately suppressing negative responses may introduce a critical side effect: gradient conflict resulting from the semantic overlap between positive and negative distributions. As highlighted in recent studies on Lazy Likelihood Displacement (LLD) (Deng et al., 2025b; 2026) and trajectory conflicts (Simoni et al., 2025), positive and negative responses often share substantial token distributions, ranging from syntactic structures to partial reasoning steps. When NSR or standard GRPO penalizes a negative trajectory, it inadvertently decrease the likelihood of shared token distributions that also occur in positive trajectories. In contrast to vanilla GRPO, this effect is amplified in NSR due to

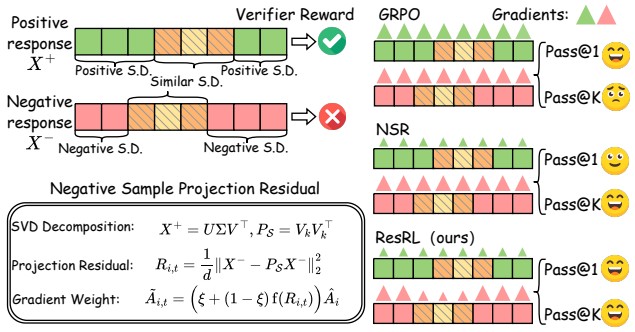

*Figure 1.* **ResRL overview.** Overlapping positive/negative semantic distributions (S.D.) can cause GRPO/NSR to penalize shared valid tokens. ResRL utilizes negative projection residuals $R_{i,t}$ to reweight gradients, reducing shared semantic penalties.

its increased negative weighting. Consequently, while NSR effectively improves Pass@$k$, it may demonstrate limited efficacy in boosting Pass@1.

This motivates a central question: **How can we disentangle the policy optimization of positive and negative responses to selectively suppress errors without penalizing the valid semantic distributions shared with correct trajectories?**

To this end, we propose ResRL to decouple the gradient updates on the overlapping regions of distributions between positive and negative responses. As shown in Figure 1, our key insight is that penalties applied to negative samples should be confined to the gradient directions orthogonal to the representations of positive samples. To operationalize this, we leverage the hidden states of the policy model as a proxy for the semantic distribution ([Zhao et al., 2025](#); [Xin et al., 2025](#)). Subsequently, we identify and selectively suppress the orthogonal complement of the negative sample's representation relative to the subspace spanned by positive ones. This mechanism ensures that shared semantic components remain preserved, while unique, erroneous reasoning patterns are targeted for suppression. To ensure computational feasibility and robustness against variations in generation length, we employ low-rank approximation to construct representation space. Extensive experiments on twelve benchmarks demonstrate that ResRL achieves state-of-the-art (SOTA) performance regarding Avg@16 (the average of 16 independent Pass@1) and Pass@128, surpassing strong baselines such as GRPO and NSR. The main contributions are summarized as follows:

- **Theoretical Framework for Gradient Decoupling:** We establish a theoretical connection between LLD and negative-positive gradient interference in NSR, proving that the inner product of output head gradients explicitly decomposes into logit and representation components. Building on this decomposition, we propose a single-forward proxy metric and theoretically

demonstrate it serves as a monotonic upper bound on representation alignment, guiding advantage reweighting to impose a conservative bound on head-gradient interference that mitigates the deleterious effects of LLD.

- **Methodological Innovation:** We present ResRL, a novel RLVR framework incorporating a semantic decoupling mechanism that leverages policy hidden states to characterize token-level response representations. By computing the residual of the negative sample's distribution after projecting onto the positive subspace, we dynamically modulate the gradient penalty during policy optimization. Furthermore, we mitigate computational overhead via a sampling-based low-rank decomposition of the positive representation matrix, complemented by a length-scaled reward mechanism that serves as a safeguard against verbosity to ensure efficient generation.

- **Empirical Performance:** We evaluate ResRL on twelve benchmarks spanning Mathematical reasoning, Code generation, Agent Tasks, and Function Calling. ResRL achieves simultaneous gains in Avg@16 and Pass@128, consistently outperforming strong baselines. On mathematics, it improves over the diversity-oriented NSR baseline by 9.4% Avg@16 on Qwen3-4B, and by 7.0% on average Pass@128. In code generation, ResRL sets a new state of the art on CodeForces, improving over NSR by 9.6% in rating. For agent tasks it outperforms EMPG on ALFWorld by 10.4% in success rate, and for function call, it exceeds ResT on multi-turn tool-use with a 2.8% gain in accuracy. Comprehensive ablation studies on factors such as rank selection, hidden layer choice, and quantile thresholds confirm that the proposed modules are synergistic and indispensable for enhancing performance.

## 2. Related Work

In recent years, RLVR has emerged as a dominant paradigm for eliciting reasoning capabilities of LLMs ([Guo et al., 2025](#)). However, debate persists regarding whether it genuinely instills novel reasoning skills or merely refines the retrieval of pre-existing patterns ([Yue et al., 2025](#); [Deng et al., 2025a](#)), often risking convergence toward spurious rewards ([Shao et al., 2025](#)). To mitigate the propensity of RLVR to prematurely narrow the search space ([Deng et al., 2025a](#)), recent studies have introduced enhanced exploration mechanisms, ranging from Monte Carlo Tree Search (MCTS) ([Wu et al., 2025](#)) to adaptive Pass@$k$ objectives ([Chen et al., 2025](#); [Yang et al., 2025b](#)). While some approaches derive closed-form gradients for Pass@$k$ ([Walder & Karkhanis, 2025](#)) or employ differentiable top-1 approximations ([Peng et al., 2025](#)), others caution that optimizing such metrics di-

rectly may induce mode collapse (Yu, 2025). Concurrently, researchers seek to refine supervision by augmenting sparse verifiers with intrinsic signals, leveraging structural proxies (Xin et al., 2025), probability divergence (Zhao et al., 2025), uncertainty estimates (Wang et al., 2025), or hidden state distributions (Zhu et al., 2025a; Deng et al., 2025c) to guide exploration in RLVR training.

Despite these advances, a critical bottleneck persists in the policy optimization: Conflicting gradients arising from semantically similar tokens across positive and negative samples (Simoni et al., 2025). This conflict frequently precipitates training instability, most notably manifesting as the LLD (Deng et al., 2025b; 2026). Although methods such as negative upweighting (Zhu et al., 2025b) and token-level loss balancing (Zeng et al., 2024) provide partial mitigation, they fail to explicitly disentangle the semantic distribution overlap between positive and negative responses. This limitation restricts their potential to robustly improve reasoning capabilities. Moreover, the strategy of utilizing projection residuals to decouple the similar semantic distribution remains unexplored, presenting an open challenge for effectively boosting both Pass@1 and Pass@$k$ metrics.

## 3. Method

### 3.1. Theoretical Framework

**Preliminaries.** Given a prompt $c$, the policy $\pi_\theta$ samples a group of $G$ trajectories $\mathcal{G} = \{y_1, \ldots, y_G\}$, where trajectory $i$ has tokens $y_{i,t}$ indexed by the time step $t \in \{1, \ldots, T_i\}$. A verifier assigns a binary trajectory-level reward $r_i \in \{0, 1\}$. GRPO optimizes the clipped policy-gradient objective with group-normalized advantages:

$$\mathcal{L}_{\text{GRPO}}(\theta) = \mathbb{E}_{c, \{y_i\}_{i=1}^G} \left[ \frac{1}{G} \sum_{i=1}^{G} \frac{1}{T_i} \sum_{t=1}^{T_i} \min \left( \rho_{i,t} \hat{A}_i, \right. \right.$$
$$\left. \left. \text{clip}(\rho_{i,t}, 1 - \epsilon, 1 + \epsilon) \hat{A}_i \right) \right], \quad (1)$$

where $\rho_{i,t} = \frac{\pi_\theta(y_{i,t}|c, y_{i,<t})}{\pi_{\theta_{\text{old}}}(y_{i,t}|c, y_{i,<t})}$ is the importance sampling ratio, and $\epsilon$ is the clipping coefficient. The advantage $\hat{A}_i$ is computed by normalizing rewards within the group. Keeping only terms with $\hat{A}_i > 0$ corresponds to positive sample reinforcement (PSR), whereas keeping only terms with $\hat{A}_i < 0$ corresponds to negative sample reinforcement (NSR).

**Theoretical Analysis.** We develop a theoretical framework that links LLD to negative–positive head-gradient interference, decomposes the output-head gradient inner product into logit and representation terms, and motivates a single-forward proxy that upper-bounds representation alignment to guide conservative advantage reweighting.

We start from LLD, which characterizes the failure of training to increase the log-likelihood of correct trajectories. For a prompt $c$ with a positive target $y^+$, define $\Delta(c) = \ln \pi_{\theta_{\text{fin}}}(y^+|c) - \ln \pi_{\theta_{\text{init}}}(y^+|c)$ as the training-induced log-likelihood gain of $y^+$. Defining $\ell = -\log \pi(\cdot)$ and assuming small output-head updates, $\Delta(c)$ admits the first-order approximation

$$\Delta(c) \approx -\eta \sum_{(i,t) \in \mathcal{N}(c)} \left\langle \nabla_W \ell^+, g_{i,t}^- \right\rangle$$
$$\propto -\eta \sum_{(i,t) \in \mathcal{N}(c)} \frac{1}{A^+} \left\langle g^+, g_{i,t}^- \right\rangle, \quad (2)$$

where $\mathcal{N}(c)$ indexes token positions $(i, t)$ from negative trajectories sampled under the same prompt $c$ that contribute to the head update, $g^+ \triangleq \nabla_W \ell^+$, and $g_{i,t}^- \triangleq \nabla_W \ell_{i,t}^-$ denote the corresponding output-head gradients (w.r.t. $W$). Here $A^+ > 0$ denotes the advantage weight of the positive trajectory. Thus, LLD is governed by accumulated cross-sign head-gradient interference.

Although gradient inner products directly quantify LLD (Yu et al., 2020), token-wise full-parameter evaluation is prohibitive at scale (extra backward passes, parameter-sized communication, and sharding-induced variance) (Rajbhandari et al., 2020) as shown in Appendix C.1. We therefore focus on the output head $W$, where gradients factorize, and use a stable single-forward geometric proxy: the orthogonal-complement energy $e(x)$.

Let $x \in \mathbb{R}^d$ denote a token representation immediately before the output head. Standard language models produce logits via a linear output head $z = Wx$, and the token loss takes the form $\ell = \ell(z)$. Under this setting, head-gradient alignment factorizes into logit and representation components, motivating representation geometry as a proxy for gradient interference.

**Lemma 1** (Gradient inner-product decomposition). *Let $\delta = \nabla_z \ell \in \mathbb{R}^{|\mathcal{V}|}$ be the backprop signal at the logits. Since $\nabla_W \ell = \delta x^\top$, for any $(\delta_1, x_1)$ and $(\delta_2, x_2)$, (Appendix A.1)*

$$\langle \nabla_W \ell_1, \nabla_W \ell_2 \rangle = \langle \delta_1, \delta_2 \rangle \cdot \langle x_1, x_2 \rangle. \quad (3)$$

With token-wise scaling $A_{i,t}$, define the effective head update $g_{i,t} \propto A_{i,t} \nabla_W \ell_{i,t}$, where $\propto$ suppresses a shared token-independent positive scalar. By Lemma 1, we get

$$|\langle g_1, g_2 \rangle| \propto |A_1 A_2| \, |\langle \delta_1, \delta_2 \rangle| \, |\langle x_1, x_2 \rangle|. \quad (4)$$

Thus, cross-sign head-gradient interference $|\langle g^-, g^+ \rangle|$ splits into a logit-space term $|\langle \delta^-, \delta^+ \rangle|$ and a representation term $|\langle x^-, x^+ \rangle|$ (Appendix A.2).

To avoid token-wise gradient estimation, we upper-bound the within-group alignment $|\langle x^-, x^+ \rangle|$, treating $|\langle \delta^-, \delta^+ \rangle|$

as an unmodeled multiplicative factor. Motivated by anisotropy and approximate low-rank structure in Transformer representations (Joshi et al., 2025; Inkiriwang et al., 2025), we fit positives with a rank-$k$ subspace.

**Definition 1** (Positive subspace construction). Let $\mathcal{P}$ be the set of positive tokens in a prompt group, and let $X^+ \in \mathbb{R}^{|\mathcal{P}| \times d}$ stack their centered representations (preprocessing in §3.2). Let $V_k \in \mathbb{R}^{d \times k}$ be the top-$k$ principal directions of $X^+$. Define (Appendix A.3)

$$S = \operatorname{span}(V_k), \qquad P_S = V_k V_k^\top. \tag{5}$$

**Definition 2** (Orthogonal-complement energy). For any representation $x$, define

$$e(x) \triangleq \frac{1}{d} \left\| (I - P_S) x \right\|_2^2. \tag{6}$$

It is the normalized squared residual of $x$ w.r.t. the positive subspace $S$ (Appendix A.4).

**Lemma 2** (Alignment bound). *For any $x^+ \in S$ and any $x \in \mathbb{R}^d$,*

$$
\begin{aligned}
\langle x, x^+ \rangle^2 &\leq \|x^+\|_2^2 \left( \|x\|_2^2 - \|(I - P_S)x\|_2^2 \right) \\
&= \|x^+\|_2^2 \left( \|x\|_2^2 - d\, e(x) \right).
\end{aligned} \tag{7}
$$

Lemma 2 shows that increasing $e(x)$ decreases an upper bound on the attainable similarity between $x$ and any positive direction in $S$. Proof is deferred to Appendix A.5.

**Theorem 1** (Residual proxies gradient alignment). *Construct $S, P_S$ as in Definition 1 and $e(x)$ as in Definition 2. For any representations $(x^-, x^+)$, we bound $|\langle x^-, x^+ \rangle|$ via $e(\cdot)$*

$$
\begin{aligned}
|\langle x^-, x^+ \rangle| &\leq \|P_S x^+\|_2 \sqrt{\|x^-\|_2^2 - d\, e(x^-)} \\
&\quad + \|x^-\|_2 \sqrt{d\, e(x^+)}.
\end{aligned} \tag{8}
$$

*Consequently, for fixed $x^+$, the subspace-dependent term is monotonically decreasing in $e(x^-)$. Assuming that $S$ sufficiently covers positive tokens (i.e., $e(x^+) \leq \varepsilon_+$), we obtain (proof in Appendix A.6)*

$$
\begin{aligned}
|\langle x^-, x^+ \rangle| &\leq \|P_S x^+\|_2 \sqrt{\|x^-\|_2^2 - d\, e(x^-)} \\
&\quad + \|x^-\|_2 \sqrt{d\, \varepsilon_+},
\end{aligned} \tag{9}
$$

*which makes $e(x^-)$ a conservative proxy for interference, up to an additive error.*

To mitigate LLD, we apply the reshaped token gradient updates in Eq. 18 into Theorem 1, yielding the conservative gradient interference upper-bounds:

$$
\begin{aligned}
|\langle \tilde{g}^-, \tilde{g}^+ \rangle| &\propto \omega^- \lambda_{pos} |A^- A^+| |\langle \delta^-, \delta^+ \rangle| |\langle x^-, x^+ \rangle| \\
&\leq \omega^- \lambda_{pos} |A^- A^+| |\langle \delta^-, \delta^+ \rangle| \Big( \\
&\quad \|P_S x^+\|_2 \sqrt{\|x^-\|_2^2 - d\, e(x^-)} + \|x^-\|_2 \sqrt{d\, e(x^+)} \Big).
\end{aligned} \tag{10}
$$

## 3.2. Algorithm Design

ResRL instantiates the representation-space proxy in Theorem 1 by estimating a positive subspace $S$ from positive samples and converting each negative token's orthogonal-complement energy $e(x)$ into a token-wise NSR weight.

**Semantic Representations and Preprocessing.** We utilize the hidden states $h_{i,t} \in \mathbb{R}^d$ from the penultimate hidden layer. While the final hidden layer directly feeds the output head, we extract representations from the preceding layer to capture high-level semantic abstractions that are less biased by the immediate token-prediction objective (Rogers et al., 2020).

To strictly align with the geometric assumptions in Definition 1, we map these raw hidden states to the analysis space via normalization and centering. For a group of positive tokens $\mathcal{P}$, we first compute the group-wise centroid of the normalized representations:

$$\mu^+ = \frac{1}{|\mathcal{P}|} \sum_{h' \in \mathcal{P}} \operatorname{LN}(h'), \tag{11}$$

where $\operatorname{LN}(\cdot)$ denotes LayerNorm (Ba et al., 2016). The centered representation $x$ for any token $h$ (used for both subspace construction and energy calculation) is then obtained by:

$$x = \operatorname{LN}(h) - \mu^+. \tag{12}$$

This centering ensures that the subspace $S$ captures the covariance structure of the positive distribution, making the orthogonal-complement energy $e(x)$ a robust metric for deviation from the "correct" reasoning trajectory.

**Subspace Estimation and Residual Computation.** While Definition 1 defines the ideal subspace $S$ using the full positive set $\mathcal{P}$, computing SVD on all tokens is computationally prohibitive for long contexts. Therefore, we employ a sampling-based approximation. For each prompt group, we uniformly sample $M$ centered positive tokens to form a reference sub-matrix $\hat{X}^+ \subset \mathbb{R}^{M \times d}$. We then perform truncated SVD on this matrix:

$$\hat{X}^+ = U \Sigma V^\top, \tag{13}$$

where $U$ and $V$ contain the left and right singular vectors, respectively, and $\Sigma$ is the diagonal matrix of singular values. We extract the top-$k$ principal directions corresponding to the largest singular values to form $V_k \in \mathbb{R}^{d \times k}$ (the first $k$ columns of $V$) and construct the projector $P_S = V_k V_k^\top$.

With this estimated subspace, we quantify the gradient interference risk for each negative token $x_{i,t}^-$. We instantiate the orthogonal-complement energy $e(x)$ as the projection residual $\mathcal{R}_{i,t}$, computed as:

$$\mathcal{R}_{i,t} \triangleq \frac{1}{d} \left\| (I - P_S) x_{i,t}^- \right\|_2^2. \tag{14}$$

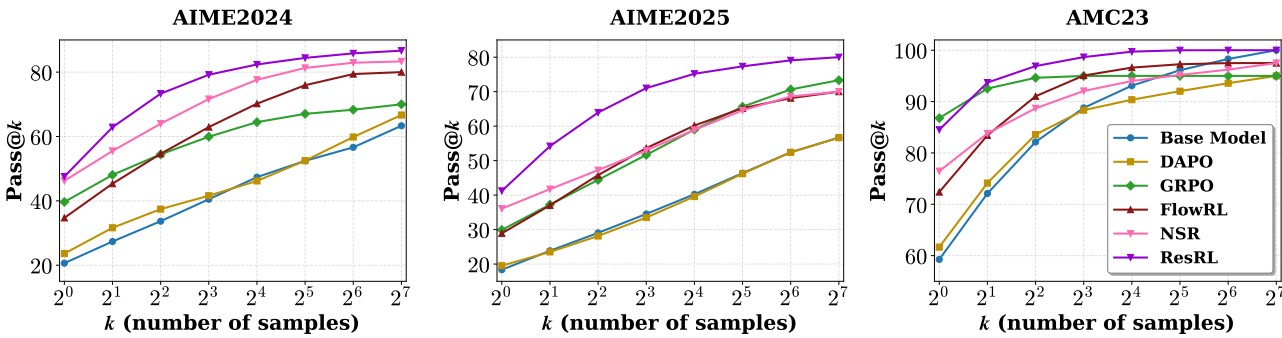

*Figure 2.* Pass@$k$ performance on AIME24/25 and AMC23 using Qwen3-4B. ResRL consistently dominates the high-$k$ regime, outperforming the base model and diversity-oriented baselines such as NSR and FlowRL, indicating a widened capability frontier.

*Table 1.* Avg@16 performance comparison on mathematical reasoning benchmarks using Qwen3 variants. All models are trained with 4096 max response length. ResRL achieves superior performance compared to existing methods.

| Method | AIME24 | AIME25 | AMC23 | MATH500 | Minerva | Olympiad | Average Acc. |
|---|---|---|---|---|---|---|---|
| Qwen3-1.7B Backbone (Yang et al., 2025a) | 11.0 | 9.8 | 43.9 | 69.5 | 26.1 | 38.1 | 33.1 |
| GRPO (Shao et al., 2024) | 12.3 | 13.8 | 54.2 | 71.5 | 27.5 | 36.0 | 35.9 |
| DAPO (Yu et al., 2025) | 10.0 | 8.4 | 57.5 | 70.9 | 30.3 | 33.6 | 35.1 |
| FlowRL(Zhu et al., 2025a) | 21.6 | 15.8 | 58.4 | 76.9 | 30.5 | 48.6 | 42.0 |
| NSR (Weighted-Reinforce) (Zhu et al., 2025b) | 27.0 | 20.4 | 66.7 | 83.5 | 33.9 | 53.5 | 47.5 |
| **ResRL (ours)** | 26.9 | 21.3 | 66.9 | 84.4 | 35.5 | 56.6 | **48.6** |
| Qwen3-4B Backbone | 20.0 | 17.3 | 56.9 | 77.8 | 36.9 | 48.2 | 42.9 |
| GRPO | 37.1 | 27.7 | 87.2 | 79.9 | 31.5 | 55.1 | 53.1 |
| DAPO | 23.5 | 18.9 | 63.4 | 80.8 | 39.1 | 51.2 | 46.2 |
| FlowRL | 35.4 | 30.2 | 74.5 | 84.7 | 38.9 | 58.1 | 53.6 |
| NSR (Weighted-Reinforce) | 38.5 | 33.1 | 79.8 | 77.4 | 33.5 | 50.1 | 52.1 |
| **ResRL (ours)** | 45.2 | 38.6 | 89.4 | 77.8 | 38.6 | 52.3 | **57.0** |
| Qwen3-8B Backbone | 25.4 | 18.1 | 61.4 | 77.6 | 39.2 | 48.6 | 45.1 |
| GRPO | 36.3 | 29.2 | 78.0 | 89.4 | 42.1 | 62.0 | 56.2 |
| DAPO | 24.2 | 24.0 | 71.3 | 76.2 | 35.3 | 43.6 | 45.8 |
| FlowRL | 47.7 | 33.3 | 85.8 | 92.1 | 44.6 | 68.5 | 62.0 |
| NSR (Weighted-Reinforce) | 55.4 | 38.5 | 89.8 | 87.3 | 40.0 | 60.6 | 61.9 |
| **ResRL (ours)** | 50.8 | 41.1 | 89.7 | 92.7 | 46.0 | 68.1 | **64.7** |

This term $\mathcal{R}_{i,t}$ serves as the tractable proxy for the theoretical interference bound derived in Theorem 1.

**Group-Relative Gating.** Since the scale of projection residuals may vary significantly across different prompts, we employ group-relative quantile normalization to robustly identify relative alignment. Let $\mathbf{D} = \{\mathcal{R}_{i,t}\}$ denote their projection residuals and $\mathcal{Q}(\mathbf{D}, \gamma)$ the empirical $\gamma$-quantile. We set

$$q_{\text{low}} = \mathcal{Q}(\mathbf{D}, \alpha), \qquad q_{\text{high}} = \mathcal{Q}(\mathbf{D}, \beta), \quad (15)$$

where $(\alpha, \beta)$ define a robust range by replacing min/max with quantiles. We then compute a quantile-based min–max normalized residual score with clipping:

$$z_{i,t} = \text{clamp}\left(\frac{\mathcal{R}_{i,t} - q_{\text{low}}}{(q_{\text{high}} - q_{\text{low}}) + \epsilon}, \, 0, \, 1\right), \quad (16)$$

*Table 2.* Performance on code reasoning benchmarks using Qwen3-4B. We report LiveCodeBench Avg/Pass@16, CodeForces Rating/Percentile (Pct.), and HumanEval+ Pass@16.

| Model | LiveCodeBench Avg/Pass@16 | CodeForces Rating (Pct.) | HumanEval+ Pass@16 |
|---|---|---|---|
| Backbone | 30.5/40.9 | 578.8 (1.2) | 89.0 |
| GRPO | 39.5/55.1 | 1267.9 (63.1) | 95.7 |
| FlowRL | 42.4/58.7 | 1333.7 (68.7) | 95.7 |
| DAPO | 41.0/52.3 | 1112.5 (46.7) | 95.7 |
| NSR | 32.8/52.3 | 1340.9 (69.3) | 96.9 |
| **ResRL** | **43.2/59.9** | **1469.5 (78.9)** | **97.0** |

where $\epsilon > 0$ prevents division by zero. Finally, we map $z_{i,t}$ to a token-wise NSR weight in $[\xi, 1]$ via

$$\omega_{i,t} = \xi + (1 - \xi)\, z_{i,t}, \quad (17)$$

where $\xi \in (0, 1]$ denotes the minimum weight.

*Table 3.* Performance comparison on ALFWorld and WebShop using Qwen2.5-7B-Instruct. We report the Success Rate (%) for ALFWorld and both Score and Success Rate for WebShop, averaged over 3 random seeds. Baseline results are adopted from (Wang et al., 2025).

| Method | ALFWorld | | | | | | | WebShop | |
|---|---|---|---|---|---|---|---|---|---|
| | Pick | Look | Clean | Heat | Cool | Pick2 | All | Task Score | Succ. |
| GPT-4o (Hurst et al., 2024) | 75.3 | 60.8 | 31.2 | 56.7 | 21.6 | 49.8 | 48.0 | 31.8 | 23.7 |
| Gemini-2.5-Pro (Comanici et al., 2025) | 92.8 | 63.3 | 62.1 | 69.0 | 26.6 | 58.7 | 60.3 | 42.5 | 35.9 |
| Prompting Backbone | 33.4 | 21.6 | 19.3 | 6.9 | 2.8 | 3.2 | 14.8 | 26.4 | 7.8 |
| Prompting ReAct (Yao et al., 2022b) | 48.5 | 35.4 | 34.3 | 13.2 | 18.2 | 17.6 | 31.2 | 46.2 | 19.5 |
| PPO (with critic) (Ouyang et al., 2022) | 92.3 | 64.0 | 92.5 | 89.5 | 80.3 | 68.8 | 80.4 | 81.4 | 68.7 |
| GRPO (Shao et al., 2024) | 88.8 | 43.7 | 88.1 | 70.3 | 77.7 | 56.8 | 74.8 | 77.8 | 65.6 |
| EMPG (Wang et al., 2025) | 92.9 | 75.2 | 74.8 | 86.3 | 73.7 | 65.3 | 78.5 | 81.0 | 69.3 |
| **ResRL (ours)** | 90.1 | 85.5 | 98.0 | 83.0 | 78.7 | 84.2 | **86.7** | **81.2** | **71.5** |

**Objective Function.** The advantages of policy optimization utilize token-wise coefficient $\tilde{A}_{i,t}$:

$$\tilde{A}_{i,t} = \begin{cases} \lambda_{pos}\hat{A}_i, & \hat{A}_i > 0, \\ \omega_{i,t}\hat{A}_i, & \hat{A}_i \leq 0. \end{cases} \quad (18)$$

For positive advantages ($\hat{A}_i > 0$), we employ a small positive scaling $\lambda_{pos} = 0.1$ as a weak anchoring mechanism to prevent model collapse following (Zhu et al., 2025b). The weight $\omega_{i,t}$ for negative samples ($\hat{A}_i \leq 0$) is defined by Eq. (17). Formally, the optimization objective of ResRL is defined as:

$$\mathcal{L}_{\text{ResRL}}(\theta) = \mathbb{E}_{x,\mathcal{G}}\left[\frac{1}{G}\sum_{i=1}^{G}\frac{1}{T_i}\sum_{t=1}^{T_i}\min\left(\rho_{i,t}\tilde{A}_{i,t},\right.\right.$$
$$\left.\left. \text{clip}(\rho_{i,t}, 1-\epsilon, 1+\epsilon)\tilde{A}_{i,t}\right)\right]. \quad (19)$$

Eq. (19) indicates that negative tokens whose representations are highly aligned with the positive subspace are down-weighted, reducing the probability of accidentally suppressing shared positive directions; tokens deviating into the orthogonal complement receive a relatively higher penalty by being assigned higher weights (Algorithm 1).

## 4. Experiment Analysis

### 4.1. Training Details

**Baselines.** We compare our method against RLVR and NSR baselines on twelve benchmarks spanning Mathematics, Code, Agent tasks, and Function Calling. These baselines include (i) GRPO (Shao et al., 2024), DAPO (Yu et al., 2025), FlowRL (Zhu et al., 2025a), and NSR (Zhu et al., 2025b) for math and code tasks; (ii) ReAct (Yao et al., 2022b), PPO (Ouyang et al., 2022), GRPO, and EMPG (Wang et al., 2025) for long-horizon agent tasks; and (iii) ResT (Lin et al., 2025a), ToolACE(Liu et al., 2025) and NSR

for function call tasks. To verify the scalability of ResRL and align with base models of these baselines, we employ several variants of the Qwen series as our base models with parameters ranging from 1.7B $\sim$ 8B.

**Training Datasets.** For mathematics, we use the DAPO training set (Yu et al., 2025) and train in `no-think` mode with a 4096-token budget. For code, we adopt the Deep-Coder dataset and train in `think` mode with an 8192-token budget. For agent tasks, we conduct experiments following the settings in (Wang et al., 2025). For function calling, we adopt the same training set as ToolRL (Qian et al., 2025; Lin et al., 2025b). Following official `veRL` (Sheng et al., 2025) implementations, we ensure fair comparison by employing identical hyperparameters, including learning rate, batch size, and training duration, while evaluating all models after training to convergence under the same budget.

**Evaluation Metrics.** We evaluate on math benchmarks (AIME 2024/2025 (MAA, 2025), AMC 2023 (MAA, 2023), MATH-500 (Lightman et al., 2023), Minerva (Lewkowycz et al., 2022), Olympiad (He et al., 2024)), code benchmarks (LiveCodeBench (Jain et al., 2024), CodeForces (Penedo et al., 2025), HumanEval+ (Chen et al., 2021)), agent benchmarks (WebShop (Yao et al., 2022a), ALFWorld (Shridhar et al., 2020)), and function calling (BFCL (Patil et al., 2024)). We report Avg@16 accuracy in Table 1 (mean over 16 independent generations), and additionally CodeForces Elo and percentile in Table 2. For math/code, we use temperature 0.6, $top\_p = 0.95$, and an 8,192 max response length (Zhu et al., 2025a); for agents, we use rollout temperature 1.0 with a 50-step cap for ALFWorld and 15 for WebShop (Wang et al., 2025).

### 4.2. Main Results

ResRL yields consistent improvements across mathematics, code, long-horizon agents, and tool-use. On Mathematical benchmarks in Table 1, ResRL indicates best per-

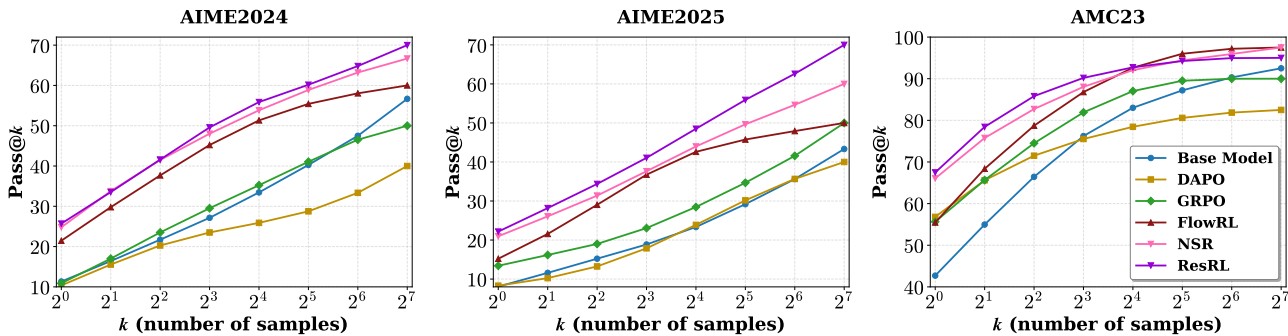

*Figure 3.* Pass@$k$ performance on AIME24/25 and AMC23 using Qwen3-1.7B. ResRL demonstrates consistent superiority on the challenging AIME datasets across all sampling budgets ($k = 2^0$ to $2^7$). On AMC23, ResRL leads in low-sample regimes and converges with baselines at high $k$ due to task saturation.

*Table 4.* Performance on BFCL benchmark. The column abbreviations stand for: **OA** (Overall), **B** (Base), **MF** (Miss Func), **MP** (Miss Param), **LC** (Long Context), **NL** (Non-Live), and **L** (Live). Baseline results are adopted from (Patil et al., 2024) and (Lin et al., 2025a).

| Models | Parameter | Multi-Turn | | | | | Single-Turn | | Overall Acc. |
|---|---|---|---|---|---|---|---|---|---|
| | | OA | B | MF | MP | LC | NL | L | |
| GPT-5-2025-08-07 | / | 28.50 | 33.50 | 29.50 | 23.00 | 28.00 | 72.92 | 58.25 | 52.65 |
| Grok-4-0709 | / | 36.12 | 44.00 | 31.00 | 26.00 | 43.50 | 85.21 | 74.39 | 64.56 |
| Qwen3-235B-A22B(Yang et al., 2025a) | 235B | 40.12 | 49.00 | 41.00 | 29.50 | 41.00 | 87.90 | 77.03 | 67.69 |
| ToolACE-2-8B(Liu et al., 2025) | 8B | 37.00 | 47.00 | 31.00 | 28.00 | 42.00 | 87.87 | 77.20 | 66.65 |
| ResT-8B (Lin et al., 2025a) | 8B | 40.13 | 50.50 | 45.00 | 32.00 | 33.00 | 90.08 | 79.03 | 68.76 |
| NSR | 8B | 36.37 | 43.00 | 41.00 | 29.00 | 32.50 | 88.00 | 80.23 | 67.80 |
| **ResRL (ours)** | 8B | **41.25** | 48.50 | 47.00 | 34.00 | 35.50 | 89.46 | 78.14 | **68.95** |

formance regarding Avg@16 and outperforms the second-best FlowRL by 15.7%, 6.3%, and 4.2% on 1.7B, 4B, and 8B, respectively. It also outperforms NSR on Avg@16 by 2.3%, 9.4%, and 4.5% on 1.7B, 4B, and 8B, indicating that semantic decoupling yields additional gains beyond negative upweighting. The improvements concentrate on harder subsets: on Qwen3-4B, ResRL boosts AIME24, AIME25, and AMC23 by 27.7%, 27.8%, and 20.0% over FlowRL; on Qwen3-8B, it increases AIME25 by 23.4% over FlowRL. We additionally compare the performance of NSR and ResRL on Qwen3-32B in Table 5. Pass@$k$ curves in Figures 2, 3, 5 further show higher low-$k$ accuracy without sacrificing high-$k$ performance; in particular, averaged over AIME24, AIME25, and AMC23 at $k=128$, our method improves Pass@128 by 7.0% over NSR on Qwen3-4B.

Importantly, these benefits extend beyond mathematics, consistent with ResRL's projection-residual reweighting that suppresses error-specific components while preserving shared prefixes. On CodeForces benchmarks in Table 2, ResRL achieves the top rating (1469.5), improving over NSR (1340.9) by 9.6%, and increases percentile by 13.9%. On ALFWorld benchmark in Table 3, it attains 86.7 overall success, surpassing PPO by 7.8% and EMPG by 10.4%. On BFCL benchmark in Table 4, ResRL delivers the best Multi-Turn OA (2.8% over ResT) and improves Miss Func

/ Miss Param by 4.4% and 6.3%.

### 4.3. Ablation Analysis

**Rank Selection.** The rank $k$ sets a protection–discrimination tradeoff: larger $k$ expands the positive subspace $S$ and reduces residual energies $\mathcal{R}_{i,t}$, but overly large $S$ can also absorb error-specific directions and weaken discrimination (consistent with the anisotropic, effectively low-rank geometry of Transformer representations (Ethayarajh, 2019; Aghajanyan et al., 2021)).

To validate this, we sweep $k$ on AIME24/25 in Figure 4. An intermediate rank ($k=64$) is both the most accurate and the most stable. With $k=8$, $S$ under-covers shared semantics, so shared-but-negative tokens are over-penalized; with $k\geq128$, residual contrast collapses for many negatives (more tokens receive small $\mathcal{R}_{i,t}$ after normalization), leading to oscillatory updates and bursty gradient norms.

**Hidden Layer Selection.** We compare using the penultimate versus the final hidden layer for representation extraction. The penultimate layer consistently achieves higher accuracy on AIME 2024/2025 in Figure 7, suggesting it provides a more stable semantic signal while being less entangled with the final layer's output-bound, next-token prediction bias. In addition, higher actor KL and entropy

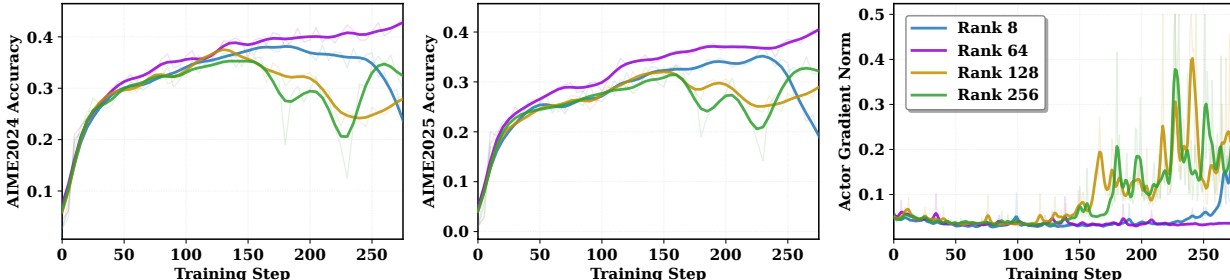

*Figure 4.* Impact of rank $k$ on model performance and optimization stability. (a) AIME2024 and (b) AIME2025 accuracy (Avg@16) curves across different ranks ($k = 8$, $k = 64$, $k = 128$, $k = 256$), demonstrating the protection-discrimination tradeoff. (c) Actor gradient norm highlighting the stability of updates, with larger ranks showing bursty gradients indicative of high variance.

indicate broader but controlled exploration, allowing the policy to refine reasoning trajectories without prematurely collapsing to suboptimal trajectories.

**Quantile Hyperparameter Selection.** We study the quantile threshold in Equation 15 by sweeping $q \in \{0.1, 0.2, 0.3\}$ in Figure 8. On AIME 2024/2025, stricter thresholds ($q$=0.1 or 0.2) converge faster and reach higher accuracy than the more permissive $q$=0.3, consistent with stronger residual-based weighting. Lower $q$ also increases actor KL and entropy, indicating broader exploration; importantly, $q$=0.1 keeps gradient variance low, achieving exploration without destabilizing optimization.

**Length-scaled Rewards.** To test long-horizon training stability without an explicit KL penalty, we train ResRL (Qwen3-8B) for 800 steps in Figure 9 and apply a length-scaled discount to positive rewards: no change up to 3500 tokens, then linearly down to 70% over 3500–4096. ResRL continues improving on AIME 2024/2025 with stable optimization (non-degenerate actor entropy and bounded, low-variance gradient norms). Meanwhile, KL increases smoothly while mean response length remains flat, suggesting the discount curbs length-based reward exploitation; overall, projection-based weighting stabilizes learning without KL, and length scaling serves as a lightweight safeguard against verbosity.

**SVD Subspace Budget.** ResRL estimates each group's positive subspace $S$ from a subsample $X^+$ of at most $M_{\max}$ positive tokens, after the normalization in Definition 1, and forms the rank-$k$ projector $P_S$. Since truncated SVD cost is correlated with $M_{\max}$, we cap $M_{\max}$ to bound overhead under long responses (4096 tokens) and grouped rollouts ($G$=4). Owing to local redundancy and low intrinsic dimensionality, the dominant directions of $X^+$ are recoverable from moderate subsamples (Zuo et al., 2025; Ethayarajh, 2019; Aghajanyan et al., 2021).

Sweeping $M_{\max} \in \{2048, 4096, 6144, 8192\}$ in Figure 10, performance is robust for moderate budgets, with dimin-

ishing returns beyond 4096. $M_{\max}$=4096 is consistently strong on AIME2024/2025 and yields stable optimization, whereas $M_{\max}$=2048 slightly lags, consistent with noisier subspace estimates and less reliable quantile-mapped weights $\omega_{i,t}$ under long responses. Increasing $M_{\max}$ further can compress residual contrast at fixed $k$, pushing $\omega_{i,t}$ toward its floor $\xi$ and weakening negative shaping (e.g., $M_{\max}$=8192 lowers KL but slows accuracy gains), while $M_{\max}$=6144 appears more susceptible to drift without accuracy benefit. We use $M_{\max}$=4096 by default.

**LayerNorm Mechanism.** We ablate the representation normalization applied before subspace projection (token-wise LayerNorm plus group-wise centering). Removing this stage sharply degrades reasoning accuracy on AIME 2024/2025 and destabilizes optimization, with high-variance gradient norms and irregular KL behavior in Figure 11. These results indicate that normalization is necessary to make residual signals comparable across tokens, preventing erratic updates and optimization collapse.

**KL Penalty Analysis.** KL regularization can stabilize GRPO but may overly constrain the exploration needed for long-horizon reasoning. In ResRL, the projection-based weight $\omega_{i,t}$ (Eq. 18) acts as an intrinsic regularizer: it attenuates negative gradients for tokens aligned with the positive subspace (low $e(x^-)$), protecting valid reasoning steps without explicitly tethering updates to the SFT prior. Removing the KL term improves AIME2024 accuracy by 9% while remaining stable in Figure 6; the KL divergence still rises, indicating controlled drift for optimizing reasoning chains rather than the destructive gradient conflicts observed in unconstrained NSR.

## 5. Conclusion

We propose ResRL, aiming to improve reasoning without sacrificing generation diversity. ResRL is motivated by a theoretical connection between LLD and negative–positive gradient interference in NSR, and introduces a single-forward

proxy metric that conservatively controls this interference via bounded representation alignment. ResRL leverages policy hidden states to represent token-level semantic distributions, constructs an efficient low-rank positive subspace via SVD, and reweights optimization using projection residuals so that negative updates primarily target error-specific components while preserving semantics shared with correct trajectories. Across twelve benchmarks spanning Mathematics, Code, Agent tasks, and Function calling, ResRL consistently improves both Pass@1 and Pass@$k$ over strong GRPO/NSR baselines while maintaining diversity; notably, it surpasses NSR on mathematical reasoning by 9.4% in Avg@16 and 7.0% in Pass@128. These results validate the efficacy and scalability of ResRL in RLVR training.

## Acknowledgments

This work was supported by the National Natural Science Foundation of China (Grant Nos. 62576338, 62550062, 62425606, and 32341009), and the Beijing Natural Science Foundation (Grant Nos. L252145 and L257008).

## Impact Statement

This paper presents work whose goal is to advance the field of LLM Reasoning. There are many potential societal consequences of our work, none of which we feel must be specifically highlighted here.

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

## A. Proofs and Derivation Details for the Theoretical Framework

### A.1. Proof of Lemma 1

Consider the linear output head that maps a token representation $x \in \mathbb{R}^d$ to logits

$$z = Wx, \qquad W \in \mathbb{R}^{|\mathcal{V}| \times d}, \quad z \in \mathbb{R}^{|\mathcal{V}|}. \tag{20}$$

Let the token-wise loss be a differentiable function of logits, $\ell = \ell(z)$, and define the backprop signal at logits

$$\delta := \nabla_z \ell \in \mathbb{R}^{|\mathcal{V}|}. \tag{21}$$

When taking inner products between two matrices of the same shape, $\langle \cdot, \cdot \rangle$ denotes the Frobenius inner product:

$$\langle A, B \rangle := \sum_{u=1}^{|\mathcal{V}|} \sum_{j=1}^{d} A_{uj} B_{uj} = \operatorname{tr}(A^\top B). \tag{22}$$

Here $\langle \cdot, \cdot \rangle$ denotes the Euclidean inner product for vectors and the Frobenius inner product for matrices. For vectors, $\langle a, b \rangle = a^\top b$ is the standard Euclidean inner product.

*Proof of Lemma 1.* We prove (i) $\nabla_W \ell = \delta x^\top$ and (ii) the factorization of the gradient inner product.

**Derivation of $\nabla_W \ell = \delta x^\top$ (entry-wise chain rule).** Write each logit coordinate explicitly:

$$z_u = (Wx)_u = \sum_{j=1}^{d} W_{uj} x_j, \qquad u \in \{1, \ldots, |\mathcal{V}|\}. \tag{23}$$

Fix an arbitrary entry $W_{ab}$ of $W$ (row $a$, column $b$). By the multivariate chain rule,

$$\frac{\partial \ell}{\partial W_{ab}} = \sum_{u=1}^{|\mathcal{V}|} \frac{\partial \ell}{\partial z_u} \cdot \frac{\partial z_u}{\partial W_{ab}} = \sum_{u=1}^{|\mathcal{V}|} \delta_u \cdot \frac{\partial}{\partial W_{ab}} \left( \sum_{j=1}^{d} W_{uj} x_j \right). \tag{24}$$

Now compute $\frac{\partial z_u}{\partial W_{ab}}$. Because $\frac{\partial W_{uj}}{\partial W_{ab}} = \mathbf{1}\{u = a\} \mathbf{1}\{j = b\}$,

$$\frac{\partial z_u}{\partial W_{ab}} = \sum_{j=1}^{d} x_j \frac{\partial W_{uj}}{\partial W_{ab}} = \sum_{j=1}^{d} x_j \mathbf{1}\{u = a\} \mathbf{1}\{j = b\} = \mathbf{1}\{u = a\} x_b. \tag{25}$$

Substituting back,

$$\frac{\partial \ell}{\partial W_{ab}} = \sum_{u=1}^{|\mathcal{V}|} \delta_u \mathbf{1}\{u = a\} x_b = \delta_a x_b. \tag{26}$$

Since this holds for all $(a, b)$, the gradient matrix satisfies $(\nabla_W \ell)_{ab} = \delta_a x_b$, hence

$$\nabla_W \ell = \delta x^\top. \tag{27}$$

**Factorization of $\langle \nabla_W \ell_1, \nabla_W \ell_2 \rangle$.** Consider two token instances producing pairs $(\delta_1, x_1)$ and $(\delta_2, x_2)$, so that

$$\nabla_W \ell_1 = \delta_1 x_1^\top, \qquad \nabla_W \ell_2 = \delta_2 x_2^\top. \tag{28}$$

Compute their Frobenius inner product directly by expanding the summation over all entries:

$$\langle \nabla_W \ell_1, \nabla_W \ell_2 \rangle = \sum_{u=1}^{|\mathcal{V}|} \sum_{j=1}^{d} (\nabla_W \ell_1)_{uj} (\nabla_W \ell_2)_{uj} = \sum_{u=1}^{|\mathcal{V}|} \sum_{j=1}^{d} (\delta_{1,u} x_{1,j})(\delta_{2,u} x_{2,j}) \tag{29}$$

$$= \sum_{u=1}^{|\mathcal{V}|} \left( \delta_{1,u} \delta_{2,u} \sum_{j=1}^{d} x_{1,j} x_{2,j} \right) = \left( \sum_{u=1}^{|\mathcal{V}|} \delta_{1,u} \delta_{2,u} \right) \left( \sum_{j=1}^{d} x_{1,j} x_{2,j} \right) \tag{30}$$

$$= \langle \delta_1, \delta_2 \rangle \cdot \langle x_1, x_2 \rangle, \tag{31}$$

which is exactly Eq. (3). This completes the proof. $\square$

## A.2. Proof of Eq. (4)

In GRPO-style objectives, each token term is weighted by a scalar coefficient (e.g., advantage, clipping-related multiplicative factors). Denote this coefficient by $A_{i,t}$. The main text defines an effective per-token head update (with proportionality absorbing objective-dependent constants)

$$g_{i,t} \; \propto \; A_{i,t} \, \nabla_W \ell_{i,t}. \tag{32}$$

We show that, for any two token instances,

$$|\langle g_1, g_2 \rangle| \; \propto \; |A_1 A_2| \, |\langle \delta_1, \delta_2 \rangle| \, |\langle x_1, x_2 \rangle|, \tag{33}$$

which is Eq. (4). The key is to combine bilinearity of inner products with Lemma 1.

*Proof of Eq. (4).* Take two token instances and suppress indices for readability:

$$(\delta_1, x_1, A_1) \quad \text{and} \quad (\delta_2, x_2, A_2),$$

where $\delta_k = \nabla_z \ell_k$ is the backprop signal at the logits and $x_k$ is the token representation feeding the output head.

**Step 1: Pull out scalar weights using bilinearity.**  Because $\langle \cdot, \cdot \rangle$ is bilinear,

$$\langle g_1, g_2 \rangle \propto \langle A_1 \nabla_W \ell_1, \; A_2 \nabla_W \ell_2 \rangle = A_1 A_2 \, \langle \nabla_W \ell_1, \nabla_W \ell_2 \rangle. \tag{34}$$

Taking absolute values yields

$$|\langle g_1, g_2 \rangle| \; \propto \; |A_1 A_2| \, |\langle \nabla_W \ell_1, \nabla_W \ell_2 \rangle|. \tag{35}$$

**Step 2: Apply Lemma 1 (exact head factorization).**  Lemma 1 states that $\nabla_W \ell_k = \delta_k x_k^\top$ and that the head-gradient inner product factorizes as

$$\langle \nabla_W \ell_1, \nabla_W \ell_2 \rangle = \langle \delta_1, \delta_2 \rangle \cdot \langle x_1, x_2 \rangle. \tag{36}$$

Substituting Eq. (36) into Eq. (35) gives

$$|\langle g_1, g_2 \rangle| \; \propto \; |A_1 A_2| \, |\langle \delta_1, \delta_2 \rangle| \, |\langle x_1, x_2 \rangle|, \tag{37}$$

which is Eq. (4).

**Consequence for cross-sign pairs in a prompt group.**  Within a prompt group, group-normalized advantages induce positive- and negative-weighted tokens. For a cross-sign pair $(x^-, x^+)$, Eq. (4) implies

$$|\langle g^-, g^+ \rangle| \; \propto \; |A^- A^+| \, |\langle \delta^-, \delta^+ \rangle| \, |\langle x^-, x^+ \rangle|.$$

Therefore, controlling the cross-sign representation similarity $|\langle x^-, x^+ \rangle|$ provides direct leverage over head-gradient interference up to the multiplicative factors $|A^- A^+|$ and $|\langle \delta^-, \delta^+ \rangle|$, motivating a single-forward proxy that upper-bounds $|\langle x^-, x^+ \rangle|$ in the main text. $\qquad \square$

### A.3. Details for Definition 1 (Positive subspace construction)

Within each prompt group, we approximate the geometry of *positive-token* representations by a low-rank subspace. This supplies a compact reference set for measuring whether a token (in particular, a negative token) aligns with dominant positive directions.

**Step 1: Token-wise LayerNorm and centering.** Let $h \in \mathbb{R}^d$ be a raw hidden state. Token-wise Layer Normalization computes per-token feature statistics

$$\mu(h) = \frac{1}{d} \sum_{j=1}^{d} h_j, \qquad \sigma^2(h) = \frac{1}{d} \sum_{j=1}^{d} \left( h_j - \mu(h) \right)^2, \tag{38}$$

and outputs

$$\mathrm{LN}(h) = \gamma \odot \frac{h - \mu(h)\mathbf{1}}{\sqrt{\sigma^2(h) + \epsilon}} + \beta, \tag{39}$$

where $\gamma, \beta \in \mathbb{R}^d$ are learned affine parameters, $\odot$ is elementwise multiplication, $\mathbf{1} \in \mathbb{R}^d$ is the all-ones vector, and $\epsilon > 0$ is a small constant. (When $\gamma, \beta$ are omitted in the main text for brevity, the construction and the subsequent linear-algebraic results remain unchanged because $\mathrm{LN}(h)$ is still a deterministic map producing a vector in $\mathbb{R}^d$.)

Given the positive-token set $\mathcal{P}$ in the same prompt group, define the positive mean

$$\mu^+ \; := \; \frac{1}{|\mathcal{P}|} \sum_{h' \in \mathcal{P}} \mathrm{LN}(h') \; \in \; \mathbb{R}^d. \tag{40}$$

For any token $h$ (positive or negative), we form the centered representation

$$\tilde{h} = \mathrm{LN}(h), \qquad x = \tilde{h} - \mu^+. \tag{41}$$

Centering ensures that the subspace we estimate from positives captures *directions of variation* among positives within the prompt group, rather than being dominated by a shared mean offset.

**Step 2: Construct the positive matrix $X^+$.** For each positive token $h \in \mathcal{P}$, compute $x = \mathrm{LN}(h) - \mu^+$ as in (41). Stack these centered positive vectors as rows to form

$$X^+ \in \mathbb{R}^{|\mathcal{P}| \times d}, \qquad X_{m:}^+ = x_m^\top, \tag{42}$$

where $x_m \in \mathbb{R}^d$ is the $m$-th centered positive representation.

**Step 3: PCA objective and equivalence to truncated SVD.** Define the (uncentered) empirical covariance of the centered positives

$$C \; := \; \frac{1}{|\mathcal{P}|} (X^+)^\top X^+ \in \mathbb{R}^{d \times d}. \tag{43}$$

A standard characterization of PCA is that the top-$k$ principal subspace solves

$$\max_{V \in \mathbb{R}^{d \times k}} \; \mathrm{tr}(V^\top C V) \quad \text{s.t.} \quad V^\top V = I_k, \tag{44}$$

i.e., it maximizes the variance captured by projecting onto $\mathrm{span}(V)$. The optimizer $V_k$ is given by the top-$k$ eigenvectors of $C$.

To connect this to the truncated SVD used in Definition 1, take an SVD of $X^+$:

$$X^+ = U\Sigma V^\top, \tag{45}$$

where $U \in \mathbb{R}^{|\mathcal{P}| \times r}, V \in \mathbb{R}^{d \times r}$ have orthonormal columns, $\Sigma \in \mathbb{R}^{r \times r}$ is diagonal with singular values, and $r = \mathrm{rank}(X^+)$. Then

$$C = \frac{1}{|\mathcal{P}|} (X^+)^\top X^+ = \frac{1}{|\mathcal{P}|} V\Sigma^2 V^\top. \tag{46}$$

Hence the eigenvectors of $C$ are exactly the right singular vectors of $X^+$, and the top-$k$ eigenvectors of $C$ correspond to the top-$k$ right singular vectors of $X^+$. Equivalently, writing the rank-$k$ truncated SVD $X^+ \approx U_k \Sigma_k V_k^\top$, the matrix $V_k \in \mathbb{R}^{d \times k}$ in Definition 1 is precisely the solution to (44).

**Step 4: Positive subspace and orthogonal projector.** Define the positive subspace

$$S = \mathrm{span}(V_k). \tag{47}$$

When $V_k$ has orthonormal columns ($V_k^\top V_k = I_k$), the matrix

$$P_S := V_k V_k^\top \in \mathbb{R}^{d \times d} \tag{48}$$

is the orthogonal projector onto $S$:

$$P_S^\top = P_S, \qquad P_S^2 = P_S. \tag{49}$$

For any $x \in \mathbb{R}^d$, the decomposition

$$x = P_S x + (I - P_S)x \tag{50}$$

splits $x$ into its component in the positive subspace and its orthogonal complement, which is the geometric basis for the orthogonal-complement energy defined next in Definition 2.

### A.4. Details for Definition 2 (Orthogonal-complement energy)

Definition 2 introduces a scalar statistic $e(x)$ that quantifies how much a token representation deviates from the positive subspace $S = \mathrm{span}(V_k)$ constructed from the positive tokens in the same prompt group (Definition 1). This appendix section formalizes the geometric meaning of $e(x)$ and records basic properties that are implicitly used later (e.g., in connecting subspace alignment to representation similarity and gradient interference bounds).

Let $V_k \in \mathbb{R}^{d \times k}$ have orthonormal columns ($V_k^\top V_k = I_k$), and let

$$P_S \; := \; V_k V_k^\top \in \mathbb{R}^{d \times d} \tag{51}$$

be the orthogonal projector onto $S = \mathrm{span}(V_k)$. Define the complementary projector

$$P_{S^\perp} \; := \; I - P_S. \tag{52}$$

For any $x \in \mathbb{R}^d$, Definition 2 is

$$e(x) \; \triangleq \; \frac{1}{d}\big\|(I - P_S)x\big\|_2^2 \; = \; \frac{1}{d}\|P_{S^\perp}x\|_2^2. \tag{53}$$

**Step 1: $P_S$ is an orthogonal projector and yields a Pythagorean decomposition.** Because $V_k^\top V_k = I_k$, $P_S$ is symmetric and idempotent:

$$P_S^\top = (V_k V_k^\top)^\top = V_k V_k^\top = P_S, \qquad P_S^2 = V_k(V_k^\top V_k)V_k^\top = V_k I_k V_k^\top = P_S. \tag{54}$$

Thus $P_S$ is the orthogonal projector onto $S$, and $P_{S^\perp} = I - P_S$ is the orthogonal projector onto $S^\perp$. In particular, for any $x$,

$$x = P_S x + P_{S^\perp} x, \qquad \langle P_S x, \; P_{S^\perp} x \rangle = 0. \tag{55}$$

The orthogonality in (55) implies the Pythagorean identity

$$\|x\|_2^2 = \|P_S x\|_2^2 + \|P_{S^\perp} x\|_2^2. \tag{56}$$

Therefore $e(x)$ is exactly the *normalized squared length* of the component of $x$ lying in $S^\perp$.

**Step 2: Nonnegativity, invariances, and an explicit coordinate form.** Since $e(x)$ is a squared norm scaled by $1/d$,

$$e(x) \geq 0, \qquad e(x) = 0 \iff (I - P_S)x = 0 \iff x \in S. \tag{57}$$

Moreover, using $P_S = V_k V_k^\top$, we can rewrite the residual norm in a form that makes the geometry explicit:

$$\|(I - P_S)x\|_2^2 = x^\top(I - P_S)^\top(I - P_S)x = x^\top(I - P_S)^2 x = x^\top(I - P_S)x \tag{58}$$
$$= x^\top x - x^\top P_S x = \|x\|_2^2 - x^\top V_k V_k^\top x = \|x\|_2^2 - \|V_k^\top x\|_2^2. \tag{59}$$

Combining (53) and (59),

$$e(x) = \frac{1}{d}\Big(\|x\|_2^2 - \|V_k^\top x\|_2^2\Big). \tag{60}$$

Interpretation: $\|V_k^\top x\|_2^2$ is the squared length captured by the top-$k$ positive directions, while the residual $\|x\|_2^2 - \|V_k^\top x\|_2^2$ measures what remains in directions orthogonal to positives.

**Step 3: Distance-to-subspace interpretation.** A key geometric fact is that orthogonal projection yields the closest point in a subspace under $\ell_2$ distance:

$$P_S x = \arg\min_{s \in S} \|x - s\|_2. \tag{61}$$

Consequently, the residual vector $(I - P_S)x$ is the displacement from $x$ to its closest point in $S$, and

$$\|(I - P_S)x\|_2 = \min_{s \in S}\|x - s\|_2, \qquad e(x) = \frac{1}{d}\Big(\min_{s \in S}\|x - s\|_2\Big)^2. \tag{62}$$

This formally justifies the main-text intuition: $e(x)$ is small precisely when $x$ lies close to $S$, and large when $x$ has a substantial component in $S^\perp$.

**Step 4: Normalized by $d$.** The factor $1/d$ in (6) makes $e(x)$ an average per-dimension squared residual. This is convenient for (i) comparability across models or layers with different hidden sizes, and (ii) keeping the magnitude of $e(x)$ stable as $d$ varies (e.g., when scaling model width). Formally, if the residual component has isotropic per-coordinate variance on the order of a constant, then $\|(I - P_S)x\|_2^2$ scales as $\Theta(d)$, while $e(x)$ remains $\Theta(1)$.

**Step 5: A useful upper bound relating residual energy to projection alignment.** Equation (60) immediately yields the bound

$$\|V_k^\top x\|_2^2 \leq \|x\|_2^2 \quad \implies \quad 0 \leq e(x) \leq \frac{1}{d}\|x\|_2^2. \tag{63}$$

When representations are LayerNormed (Definition 1), $\|x\|_2^2$ tends to be better controlled, making $e(x)$ a stable scalar summary of "deviation from positives" within a prompt group.

## A.5. Proof of Lemma 2 (Alignment bound)

For any $x^+ \in S$ and any $x \in \mathbb{R}^d$,

$$\langle x, x^+ \rangle^2 \leq \|x^+\|_2^2 \left( \|x\|_2^2 - \|(I - P_S)x\|_2^2 \right)$$
$$= \|x^+\|_2^2 \left( \|x\|_2^2 - d\,e(x) \right). \tag{64}$$

Let $S = \text{span}(V_k) \subseteq \mathbb{R}^d$ be the positive subspace from Definition 1. Let $P_S$ be the orthogonal projector onto $S$ (so $P_S = V_k V_k^\top$ with $V_k^\top V_k = I_k$), and let $P_{S^\perp} = I - P_S$ be the orthogonal projector onto the orthogonal complement $S^\perp$. For any $x \in \mathbb{R}^d$, define the orthogonal decomposition

$$x = x_S + x_\perp, \qquad x_S := P_S x \in S, \quad x_\perp := (I - P_S)x \in S^\perp. \tag{65}$$

By properties of orthogonal projections, $x_S \perp x_\perp$ and

$$\|x\|_2^2 = \|x_S\|_2^2 + \|x_\perp\|_2^2. \tag{66}$$

*Proof of Lemma 2.* Fix any $x^+ \in S$ and any $x \in \mathbb{R}^d$.

**Step 1: Reduce $\langle x, x^+ \rangle$ to the in-subspace component of $x$.** Using the decomposition (65) and linearity of the inner product,

$$\langle x, x^+ \rangle = \langle x_S + x_\perp,\ x^+ \rangle = \langle x_S,\ x^+ \rangle + \langle x_\perp,\ x^+ \rangle. \tag{67}$$

Since $x_\perp \in S^\perp$ and $x^+ \in S$, we have $\langle x_\perp, x^+ \rangle = 0$. Therefore,

$$\langle x, x^+ \rangle = \langle x_S, x^+ \rangle = \langle P_S x,\ x^+ \rangle. \tag{68}$$

**Step 2: Apply Cauchy–Schwarz within the subspace.** By Cauchy–Schwarz in $\mathbb{R}^d$,

$$\langle x_S, x^+ \rangle^2 \leq \|x_S\|_2^2 \, \|x^+\|_2^2. \tag{69}$$

Combining (68) and (69) yields

$$\langle x, x^+ \rangle^2 \leq \|x^+\|_2^2 \, \|P_S x\|_2^2. \tag{70}$$

**Step 3: Rewrite $\|P_S x\|_2^2$ using the residual $\|(I - P_S)x\|_2^2$.** From the Pythagorean identity (66),

$$\|x\|_2^2 = \|P_S x\|_2^2 + \|(I - P_S)x\|_2^2 \quad \implies \quad \|P_S x\|_2^2 = \|x\|_2^2 - \|(I - P_S)x\|_2^2. \tag{71}$$

Substituting (71) into (70) gives

$$\langle x, x^+ \rangle^2 \leq \|x^+\|_2^2 \left( \|x\|_2^2 - \|(I - P_S)x\|_2^2 \right), \tag{72}$$

which is the first line of (64).

**Step 4: Express the bound via the residual energy $e(x)$.** By Definition 2, $e(x) = \frac{1}{d}\|(I - P_S)x\|_2^2$, equivalently

$$\|(I - P_S)x\|_2^2 = d\,e(x). \tag{73}$$

Substituting into the previous inequality yields the second line of (64). This completes the proof. $\square$

The bound is tight: equality holds whenever $x_\perp = 0$ (i.e., $x \in S$) and $x_S$ is colinear with $x^+$. Geometrically, the lemma states that alignment with any positive direction $x^+ \in S$ is controlled by the amount of energy of $x$ that lies *inside* $S$; the residual energy in $S^\perp$ (equivalently $e(x)$) subtracts from the maximum achievable squared inner product.

**A.6. Proof of Theorem 1 (Residual proxies gradient alignment)**

A token representation $x \in \mathbb{R}^d$ immediately before the output head produces logits via a linear map $z = Wx$ (possibly with tied weights), and the token loss is $\ell = \ell(z)$. Let $\delta := \nabla_z \ell \in \mathbb{R}^{|\mathcal{V}|}$ denote the backprop signal at the logits. In GRPO-style objectives, each token term is multiplied by a scalar coefficient (advantage, clipping-induced factor, etc.); we denote this coefficient by $A_{i,t}$ and write the effective per-token head gradient as

$$g_{i,t} \propto A_{i,t} \nabla_W \ell_{i,t}. \tag{74}$$

For a prompt group, we construct the positive subspace $S = \mathrm{span}(V_k)$ and its orthogonal projector $P_S = V_k V_k^\top$ (Definition 1), and define the orthogonal-complement energy $e(x) := \frac{1}{d}\|(I - P_S)x\|_2^2$ (Definition 2). Standard facts used below include: (i) properties of orthogonal projections and Pythagorean decompositions, and (ii) Cauchy–Schwarz and the triangle inequality; see (Horn & Johnson, 2012) for projection geometry in Euclidean spaces.

*Proof of Theorem 1.* Fix any negative/positive token pair $(x^-, x^+)$ within the same prompt group, with corresponding coefficients $(A^-, A^+)$ and logit-space signals $(\delta^-, \delta^+)$.

**Step 1: exact head-gradient factorization.** By Lemma 1 (Gradient inner-product decomposition), for each token $\nabla_W \ell = \delta x^\top$ and

$$\langle \nabla_W \ell^-, \nabla_W \ell^+ \rangle = \langle \delta^-, \delta^+ \rangle \cdot \langle x^-, x^+ \rangle. \tag{75}$$

Using $g^\pm \propto A^\pm \nabla_W \ell^\pm$ and bilinearity of the inner product,

$$\langle g^-, g^+ \rangle \propto A^- A^+ \langle \nabla_W \ell^-, \nabla_W \ell^+ \rangle = A^- A^+ \langle \delta^-, \delta^+ \rangle \langle x^-, x^+ \rangle. \tag{76}$$

Taking absolute values yields

$$|\langle g^-, g^+ \rangle| \propto |A^- A^+| \, |\langle \delta^-, \delta^+ \rangle| \, |\langle x^-, x^+ \rangle|. \tag{77}$$

**Step 2: bounding $|\langle x^-, x^+ \rangle|$ by residual energies (Eq. (8)).** Define the decomposition of the positive token representation into components parallel and orthogonal to $S$:

$$x_\parallel^+ := P_S x^+ \in S, \qquad x_\perp^+ := (I - P_S)x^+ \in S^\perp, \qquad x^+ = x_\parallel^+ + x_\perp^+. \tag{78}$$

Using linearity of the inner product,

$$\langle x^-, x^+ \rangle = \langle x^-, x_\parallel^+ \rangle + \langle x^-, x_\perp^+ \rangle. \tag{79}$$

Applying the triangle inequality yields

$$|\langle x^-, x^+ \rangle| \leq |\langle x^-, x_\parallel^+ \rangle| + |\langle x^-, x_\perp^+ \rangle|. \tag{80}$$

We bound the two terms in (80) separately.

**(a) Subspace-alignment term $|\langle x^-, x_\parallel^+ \rangle|$.** Since $x_\parallel^+ \in S$ by construction, we may apply Lemma 2 (Alignment bound) with $x = x^-$ and $x^+ = x_\parallel^+$:

$$\langle x^-, x_\parallel^+ \rangle^2 \leq \|x_\parallel^+\|_2^2 \left( \|x^-\|_2^2 - \|(I - P_S)x^-\|_2^2 \right). \tag{81}$$

The right-hand side is nonnegative because orthogonal projection cannot increase norm: $\|(I - P_S)x^-\|_2^2 \leq \|x^-\|_2^2$ (a standard property of orthogonal projectors; see (Horn & Johnson, 2012)). Taking square roots on both sides of (81) gives

$$|\langle x^-, x_\parallel^+ \rangle| \leq \|x_\parallel^+\|_2 \sqrt{\|x^-\|_2^2 - \|(I - P_S)x^-\|_2^2}. \tag{82}$$

Finally, by Definition 2, $\|(I - P_S)x^-\|_2^2 = d\,e(x^-)$, so

$$|\langle x^-, x_\parallel^+ \rangle| \leq \|x_\parallel^+\|_2 \sqrt{\|x^-\|_2^2 - d\,e(x^-)}. \tag{83}$$

**(b) Orthogonal-residual term** $|\langle x^-, x_\perp^+ \rangle|$. Apply Cauchy–Schwarz in $\mathbb{R}^d$:

$$|\langle x^-, x_\perp^+ \rangle| \le \|x^-\|_2 \|x_\perp^+\|_2. \tag{84}$$

Using $x_\perp^+ = (I - P_S)x^+$ and Definition 2, we have

$$\|x_\perp^+\|_2 = \|(I - P_S)x^+\|_2 = \sqrt{\|(I - P_S)x^+\|_2^2} = \sqrt{d\,e(x^+)}. \tag{85}$$

Substituting (85) into (84) yields

$$|\langle x^-, x_\perp^+ \rangle| \le \|x^-\|_2 \sqrt{d\,e(x^+)}. \tag{86}$$

**(c) Combine (a) and (b).** Plugging (83) and (86) into (80) gives

$$|\langle x^-, x^+ \rangle| \le \|x_\parallel^+\|_2 \sqrt{\|x^-\|_2^2 - d\,e(x^-)} \;+\; \|x^-\|_2 \sqrt{d\,e(x^+)}, \tag{87}$$

which is Eq. (8).

**Step 3: monotonicity in $e(x^-)$ for fixed $x^+$.** Fix $x^+$ (hence $x_\parallel^+ = P_S x^+$ is fixed). Consider the first term on the right-hand side of (87):

$$T(e(x^-)) := \|x_\parallel^+\|_2 \sqrt{\|x^-\|_2^2 - d\,e(x^-)}. \tag{88}$$

Because $d\,e(x^-) = \|(I - P_S)x^-\|_2^2 \in [0, \|x^-\|_2^2]$, the quantity under the square root lies in $[0, \|x^-\|_2^2]$. Moreover, the map $u \mapsto \sqrt{\|x^-\|_2^2 - u}$ is strictly decreasing on $u \in [0, \|x^-\|_2^2]$, so $T(e(x^-))$ is monotonically non-increasing in $e(x^-)$.

**Step 4: a conservative proxy under positive-subspace capture.** Assume positives are well captured by $S$ such that $e(x^+) \le \varepsilon_+$ holds for most positive tokens. Then $\sqrt{d\,e(x^+)} \le \sqrt{d\,\varepsilon_+}$, and (87) implies

$$|\langle x^-, x^+ \rangle| \le \|x_\parallel^+\|_2 \sqrt{\|x^-\|_2^2 - d\,e(x^-)} \;+\; \|x^-\|_2 \sqrt{d\,\varepsilon_+}. \tag{89}$$

The first term is the $e(x^-)$-dependent (monotonically decreasing) subspace-alignment term, while the second term is an additive approximation error that depends only on how well positives are captured by $S$. Combining this inequality with Eq. (77) shows that $e(x^-)$ serves as a conservative proxy for worst-case gradient interference up to the additive error induced by imperfect positive-subspace capture, and up to the multiplicative logit-space similarity factor $|\langle \delta^-, \delta^+ \rangle|$. $\qquad\square$

## B. Algorithm Design

---

**Algorithm 1** ResRL (per prompt group)

---

**Require:** Prompt-group trajectories $\{y_i\}_{i=1}^G$ with lengths $\{T_i\}$ and tokens $y_{i,t}$; group-normalized advantages $\{\hat{A}_i\}$; penultimate-layer hidden states $\{h_{i,t} \in \mathbb{R}^d\}$; validity mask $m_{i,t} \in \{0,1\}$ (1 for non-padding tokens); rank $k$; positive-token budget $M_{\max}$; quantiles $(\alpha, \beta)$ with $0 < \alpha < \beta < 1$; negative penalty floor $\xi \in (0,1)$; stabilizer $\varepsilon > 0$; positive scaling $\lambda_+ > 0$; (optional) truncation-tail mask $\tau_{i,t} \in \{0,1\}$.

**Ensure:** Token-wise coefficients $\{\tilde{A}_{i,t}\}$ for optimizing $\mathcal{L}_{\text{ResRL}}$.

1: Split rollouts: $\mathcal{P} \leftarrow \{i : \hat{A}_i > 0\}, \mathcal{N} \leftarrow \{i : \hat{A}_i \le 0\}$.
2: **if** $|\mathcal{P}| = 0$ **then**
3:     **for** each valid token $(i,t)$ with $m_{i,t} = 1$ **do**
4:         $\tilde{A}_{i,t} \leftarrow \hat{A}_i$.
5:     **end for**
6:     Optimize the baseline clipped GRPO objective using $\tilde{A}_{i,t}$ (no reweighting).
7:     **return**
8: **end if**
9: $\mathcal{I}^+ \leftarrow \text{UNIFORMSAMPLE}(\{(i,t) : i \in \mathcal{P}, m_{i,t} = 1\}, M_{\max}); M \leftarrow |\mathcal{I}^+|$.
10: Compute LayerNormed positives $\tilde{h}_{i,t} \leftarrow \text{LN}(h_{i,t})$ for all $(i,t) \in \mathcal{I}^+$.
11: Positive mean: $\mu^+ \leftarrow \frac{1}{M} \sum_{(i,t) \in \mathcal{I}^+} \tilde{h}_{i,t}$.
12: Form $X^+ \in \mathbb{R}^{M \times d}$ with rows $x_{i,t}^+ \leftarrow \tilde{h}_{i,t} - \mu^+$.
13: Compute top-$k$ principal directions (rank-$k$ truncated SVD / PCA) of $X^+$: $V_k \in \mathbb{R}^{d \times k}$ with $V_k^\top V_k = I_k$; set projector $P_S \leftarrow V_k V_k^\top$.
14: **for** each negative token $(i,t)$ with $i \in \mathcal{N}$ and $m_{i,t} = 1$ **do**
15:     $\tilde{h}_{i,t} \leftarrow \text{LN}(h_{i,t}); x_{i,t}^- \leftarrow \tilde{h}_{i,t} - \mu^+$.
16:     Residual energy $R_{i,t} \leftarrow \frac{1}{d} \left\| (I - P_S) x_{i,t}^- \right\|_2^2$.
17: **end for**
18: Collect residuals $\mathcal{D} \leftarrow \{R_{i,t} : i \in \mathcal{N}, m_{i,t} = 1\}$.
19: Quantiles: $q_{\text{low}} \leftarrow \mathcal{Q}(\mathcal{D}, \alpha), q_{\text{high}} \leftarrow \mathcal{Q}(\mathcal{D}, \beta)$.
20: **for** each negative token $(i,t)$ with $i \in \mathcal{N}$ and $m_{i,t} = 1$ **do**
21:     $z_{i,t} \leftarrow \text{clamp}\left( \frac{R_{i,t} - q_{\text{low}}}{(q_{\text{high}} - q_{\text{low}}) + \varepsilon}, 0, 1 \right)$.
22:     $\omega_{i,t} \leftarrow \xi + (1 - \xi) z_{i,t}$.
23:     **if** truncation guard enabled and $\tau_{i,t} = 1$ **then**
24:         $\omega_{i,t} \leftarrow 1$.
25:     **end if**
26: **end for**
27: **for** each valid token $(i,t)$ with $m_{i,t} = 1$ **do**
28:     **if** $\hat{A}_i > 0$ **then**
29:         $\tilde{A}_{i,t} \leftarrow \lambda_+ \hat{A}_i$
30:     **else**
31:         $\tilde{A}_{i,t} \leftarrow \omega_{i,t} \hat{A}_i$.
32:     **end if**
33: **end for**
34: Optimize the clipped GRPO objective with $\hat{A}_i$ replaced by $\tilde{A}_{i,t}$, plus KL penalty if used.

---

## C. Time Complexity of Gradient-Inner-Product Modules

This appendix analyzes the per-prompt-group time complexity of the gradient-inner-product based modules used in RESRL (ours) and in LLD/NTHR. Throughout, we use standard Big-$\mathcal{O}$ notation and count floating-point operations up to constant factors.

**Common notation.** A prompt group consists of $G$ sampled trajectories $\{y_i\}_{i=1}^G$ with lengths $\{T_i\}$ and tokens $y_{i,t}$. We denote the penultimate-layer hidden state at token $(i,t)$ by $h_{i,t} \in \mathbb{R}^d$, where $d$ is the hidden size, and use a validity mask $m_{i,t} \in \{0,1\}$ to ignore padding tokens. Let $W \in \mathbb{R}^{|V| \times d}$ be the (token) unembedding matrix and $|V|$ be the vocabulary size.

### C.1. ResRL: Residual-based proxy for head-gradient interference

**Module overview.** RESRL constructs, per prompt group, a rank-$k$ positive subspace from (a subsample of) positive tokens and then computes the projection residual energy $R_{i,t}$ for each negative token to produce token-wise weights. We analyze the additional overhead beyond the baseline GRPO forward/backward passes.

**Step 1: forming the positive matrix.** Let $P = \{i : \widehat{A}_i > 0\}$ be the set of positive trajectories and let $I^+$ be the sampled positive token indices with $M := |I^+| \le M_{\max}$. After LayerNorm and group-wise centering, the method forms $X^+ \in \mathbb{R}^{M \times d}$ by stacking $M$ centered positive vectors. This costs $\mathcal{O}(Md)$ time and $\mathcal{O}(Md)$ memory.

**Step 2: rank-$k$ truncated SVD / PCA.** Computing the top-$k$ principal directions of $X^+$ (equivalently, the rank-$k$ truncated SVD/PCA) costs

$$\mathcal{O}(Mdk) \tag{90}$$

time using standard iterative methods (e.g., Lanczos / randomized SVD), and stores $V_k \in \mathbb{R}^{d \times k}$ with $\mathcal{O}(dk)$ memory. (A full SVD would be higher order and is unnecessary here.)

**Step 3: residual energies for negative tokens.** Let $N = \{i : \widehat{A}_i \le 0\}$ be the set of negative trajectories and let

$$T_- := \big|\{(i,t) : i \in N,\ m_{i,t} = 1\}\big|$$

be the number of valid negative tokens in the group. For each negative token, RESRL computes the centered vector $x_{i,t}^- \in \mathbb{R}^d$ and the residual energy

$$R_{i,t} = \frac{1}{d} \big\|(I - P_S)\, x_{i,t}^-\big\|_2^2, \qquad P_S := V_k V_k^\top.$$

Applying $P_S$ to a vector can be implemented as $V_k(V_k^\top x)$, which costs $\mathcal{O}(dk)$ per token. Thus computing $\{R_{i,t}\}$ costs

$$\mathcal{O}(T_- dk). \tag{91}$$

**Step 4: quantiles and weight mapping.** Let $D = \{R_{i,t} : i \in N,\ m_{i,t} = 1\}$ be the multiset of residuals. Computing the $\alpha$- and $\beta$-quantiles can be done in expected linear time via selection: $\mathcal{O}(T_-)$ (or $\mathcal{O}(T_- \log T_-)$ if implemented by sorting). The subsequent per-token mapping (clamp and affine transform) costs $\mathcal{O}(T_-)$.

**Total per-group overhead (ResRL).** Combining the above, the additional time cost per prompt group is

$$\boxed{\mathcal{O}\big(Mdk + T_- dk + T_-\big) = \mathcal{O}\big((M + T_-)dk\big), \quad \text{with } M \le M_{\max}.} \tag{92}$$

The extra memory is dominated by storing $X^+$ and $V_k$, i.e.,

$$\boxed{\mathcal{O}(Md + dk).} \tag{93}$$

In practice, $M_{\max}$ caps the SVD cost and makes the overhead predictable under long rollouts.

## C.2. LLD/NTHR: Gradient-inner-product score and efficiency tricks

**Module overview.** LLD analyzes the impact of negative gradients through a group-weighted hidden embedding score that aggregates (hidden-state) inner products between positive and negative tokens, weighted by token-level prediction-error similarity. A direct implementation of pairwise hidden-state inner products across all positive/negative token pairs has cost

$$\mathcal{O}\Big( \big( \sum_{i\in[N^+]} |y_i^+| \big) \big( \sum_{j\in[N^-]} |y_j^-| \big) d \Big), \tag{94}$$

which is quadratic in the total group token count.

**Reformulation as a matrix inner product (summations first).** LLD/NTHR notes that the score can be rewritten so that summations over tokens are computed before the final inner product, reducing redundant work. Concretely, each token contributes an outer product between a prediction-error vector (e.g., $e_y - \pi(\cdot|\cdot) \in \mathbb{R}^{|V|}$) and a hidden embedding $h \in \mathbb{R}^d$, which naively costs $\mathcal{O}(|V|d)$ per token.

**Restricting to the response vocabulary.** Since the probability mass is concentrated on tokens appearing in the generated responses, LLD/NTHR restricts computation to a response-specific vocabulary $V_x^\star$ for each prompt $x$, with $|V_x^\star| \ll |V|$. This reduces the per-token outer-product accumulation from $\mathcal{O}(|V|d)$ to

$$\mathcal{O}(|V_x^\star|d). \tag{95}$$

**Total per-group overhead (LLD/NTHR).** Let $T := \sum_{i=1}^G |y_i|$ be the total number of tokens in the group. With the above reformulation and vocabulary restriction, the dominant cost becomes linear in $T$:

$$\boxed{\mathcal{O}\big( T\,|V_x^\star|\,d \big) \quad (\text{or } \mathcal{O}(T\,|V|\,d) \text{ without restriction}).} \tag{96}$$

The final matrix inner product adds at most $\mathcal{O}(|V_x^\star|d)$, which is lower order compared to the token accumulation term. The additional memory is $\mathcal{O}(|V_x^\star|d)$ for storing the accumulated statistics.

**Takeaway.** Both methods exploit structure implied by gradient-inner-product decompositions: RESRL reduces the problem to low-rank projection in $\mathbb{R}^d$ (hence $\mathcal{O}((M + T_-)dk)$), whereas LLD/NTHR reduces quadratic token-pair interactions to a linear-time accumulation over tokens (hence $\mathcal{O}(T\,|V_x^\star|d)$ after restricting the vocabulary).

**ResRL vs. LLD/NTHR: time-complexity reduction.** Comparing the dominant per-group overhead terms, RESRL costs $\mathcal{O}((M + T_-)dk)$, while LLD/NTHR costs $\mathcal{O}(T\,|V_x^\star|\,d)$ after vocabulary restriction. Therefore, the asymptotic reduction factor in time (LLD over ResRL) is

$$\frac{\mathcal{O}(T\,|V_x^\star|\,d)}{\mathcal{O}((M + T_-)dk)} \;=\; \mathcal{O}\Big( \frac{T\,|V_x^\star|}{(M + T_-)\,k} \Big). \tag{97}$$

When $M \ll T_-$ and $T \asymp T_-$ (typical long-rollout groups), this simplifies to $\mathcal{O}(|V_x^\star|/k)$, i.e., RESRL reduces the overhead by roughly a factor of $|V_x^\star|/k$ relative to LLD/NTHR. In contrast, against a naïve LLD implementation without reformulation (quadratic token-pair cost), RESRL replaces $\mathcal{O}(T_+T_-d)$ with $\mathcal{O}((M + T_-)dk)$, yielding a much larger reduction of $\mathcal{O}\Big( \frac{T_+T_-}{(M+T_-)k} \Big)$.

# D. Additional Implementation Details

## D.1. Additional Experiments

*Table 5.* Avg@16 performance comparison on mathematical reasoning benchmarks using Qwen3-32B Base model. Models are trained with 8192 max response length.

| Method | AIME24 | AIME25 | AMC23 | MATH500 | Minerva | Olympiad | Average Acc. |
|---|---|---|---|---|---|---|---|
| **RLVR from the Qwen3-32B Base Model (Think mode, 8192 max tokens)** | | | | | | | |
| NSR (Weighted-Reinforce) | 54.7 | 45.6 | 85.8 | 88.1 | 47.7 | 64.4 | 64.4 |
| **ResRL (ours)** | 60.9 | 44.4 | 89.6 | 94.5 | 49.6 | 70.7 | **68.3** |

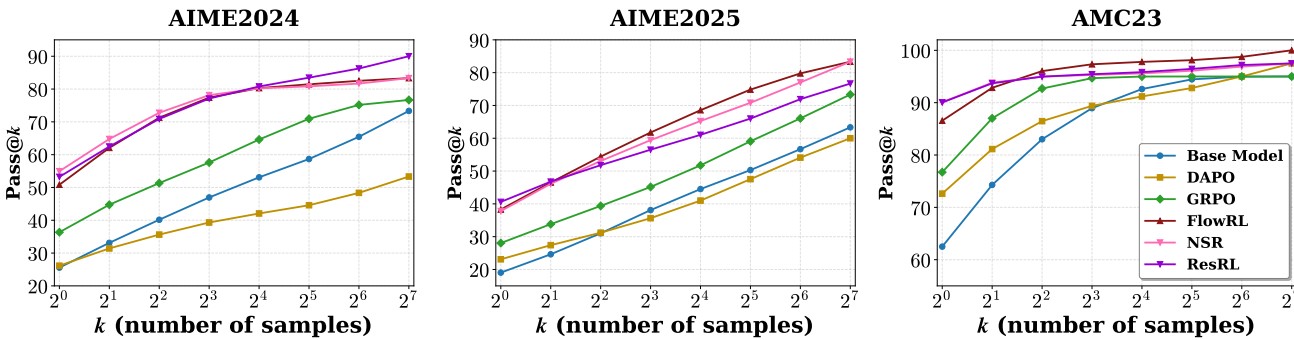

*Figure 5.* Pass@$k$ performance on AIME24/25 and AMC23 using Qwen3-8B. ResRL dominates practical low-to-mid regimes ($k \leq 2^6$) and remains competitive at high compute ($k=2^7$). This confirms ResRL optimizes the precision–diversity trade-off, securing reliable reasoning without relying on the high-variance "brute-force" exploration of unconstrained methods.

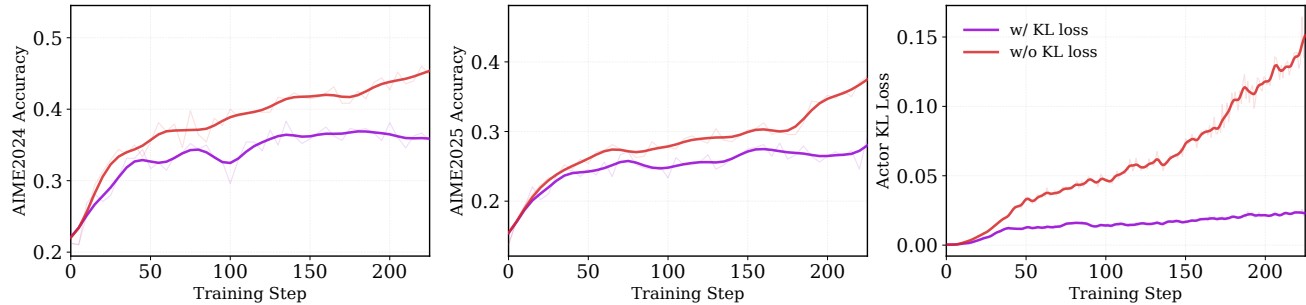

*Figure 6.* Ablation of the KL penalty using Qwen3-8B. Removing the explicit KL term (red) significantly boosts accuracy on **(a)** AIME2024 and **(b)** AIME2025 (Avg@16) compared to the standard configuration (purple). **(c)** The rising KL divergence reflects an expanded exploration horizon enabled by ResRL. Crucially, training remains stable despite this drift, confirming that ResRL's projection-based weighting acts as a sufficient intrinsic regularizer, rendering the strict SFT constraint redundant.

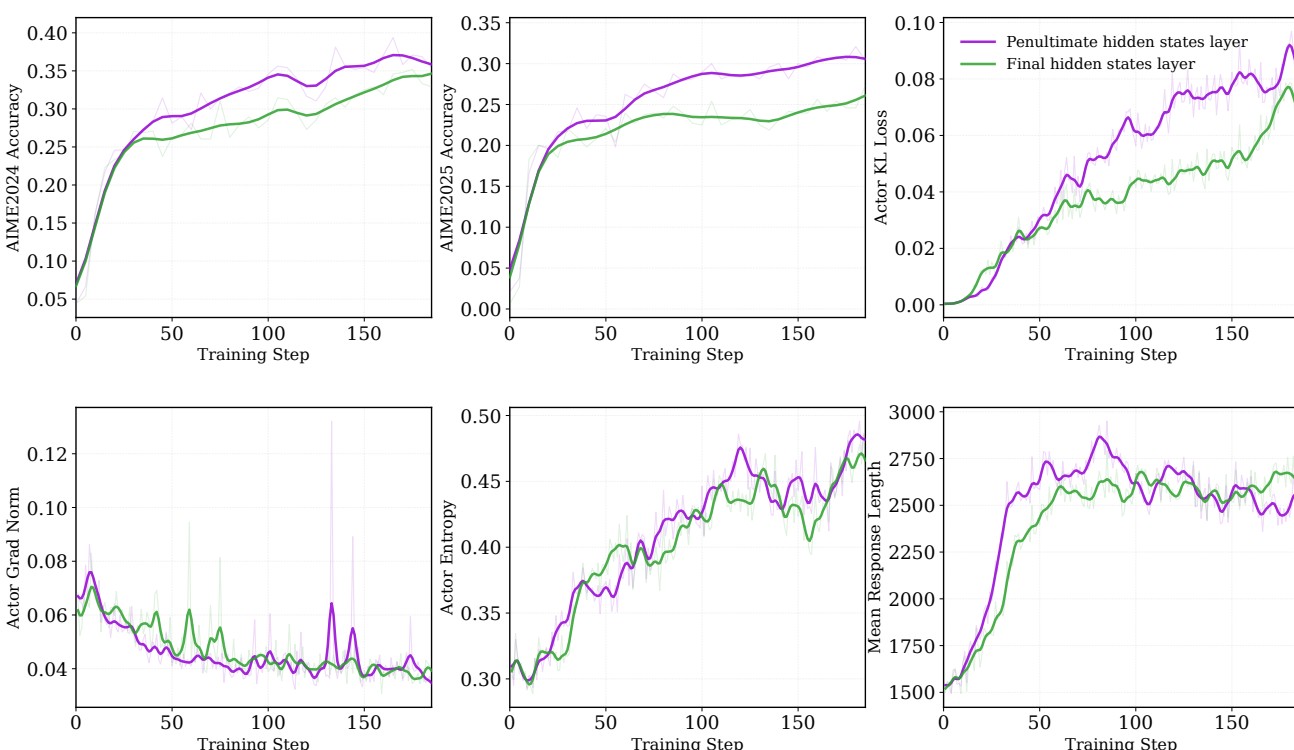

*Figure 7.* Impact of hidden layer selection on reasoning performance. Utilizing the penultimate hidden layer (purple) yields significantly superior accuracy on **(a)** AIME2024 and **(b)** AIME2025 compared to the final hidden layer (green). **(c)** The optimization dynamics, characterized by elevated KL divergence and actor entropy, indicate that the penultimate layer facilitates more sufficient exploration. Crucially, this confirms that the penultimate layer captures high-level semantic abstractions while mitigating the immediate token-prediction bias inherent to the final layer, thereby preventing premature convergence to suboptimal reasoning paths.

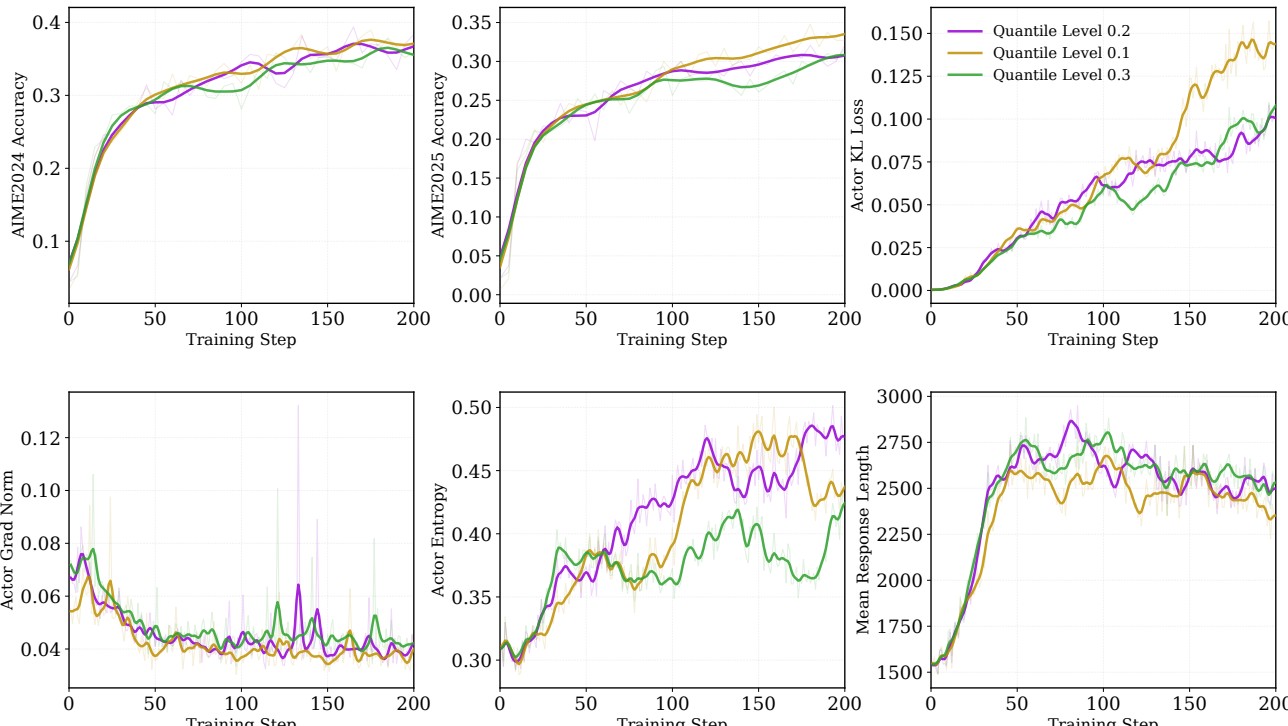

*Figure 8.* Sensitivity analysis of the quantile hyperparameter. Lower quantile thresholds (0.1 and 0.2) accelerate convergence and yield superior accuracy on **(a)** AIME2024 and **(b)** AIME2025 compared to the more permissive 0.3 level (green). **(c)** The elevated KL divergence at lower quantiles indicates that stricter residual penalization drives more aggressive exploration. Crucially, the 0.1 configuration maintains optimization stability (low gradient variance) despite this exploration, validating that the projection residual is a high-fidelity signal for error suppression, thus justifying a stricter gating mechanism.

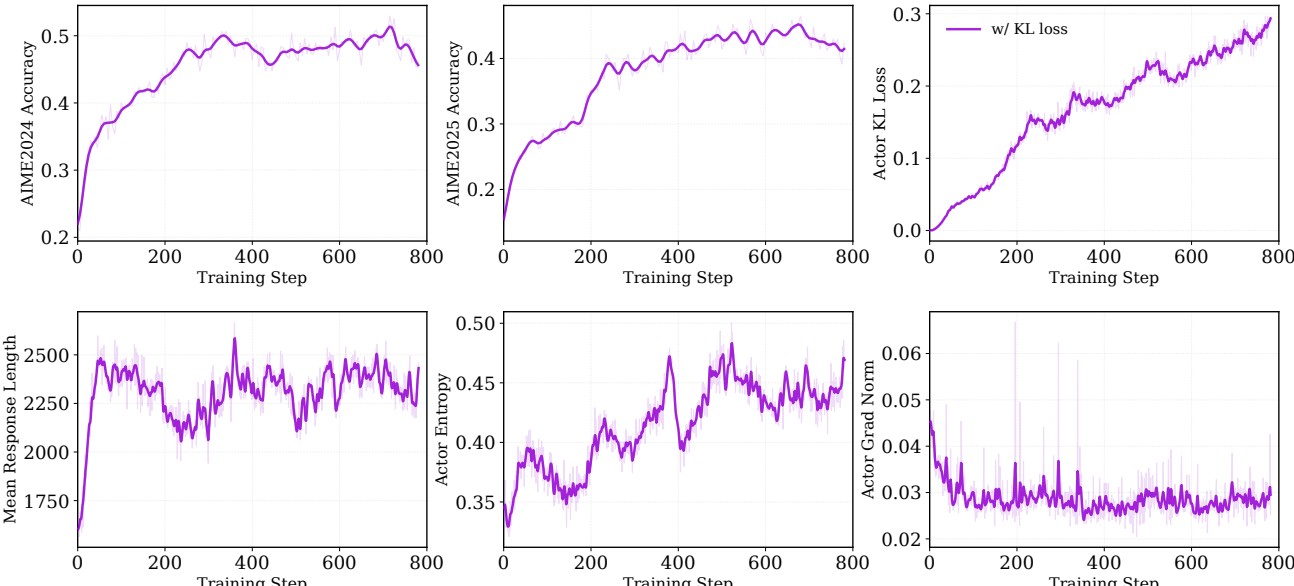

*Figure 9.* **Long-horizon training stability of ResRL (Qwen3-8B) with explicit KL regularization.** We extend the training to 800 steps to verify asymptotic stability. Despite the removal of the KL penalty, the model exhibits *(Top)* continuous performance gains on AIME 2024/2025 and a natural, bounded rise in KL divergence indicative of effective exploration; and *(Bottom)* stable dynamics in response length, actor entropy, and gradient norms. This confirms that ResRL's subspace-based semantic constraints successfully prevent mode collapse and reward hacking without requiring rigid SFT anchoring.

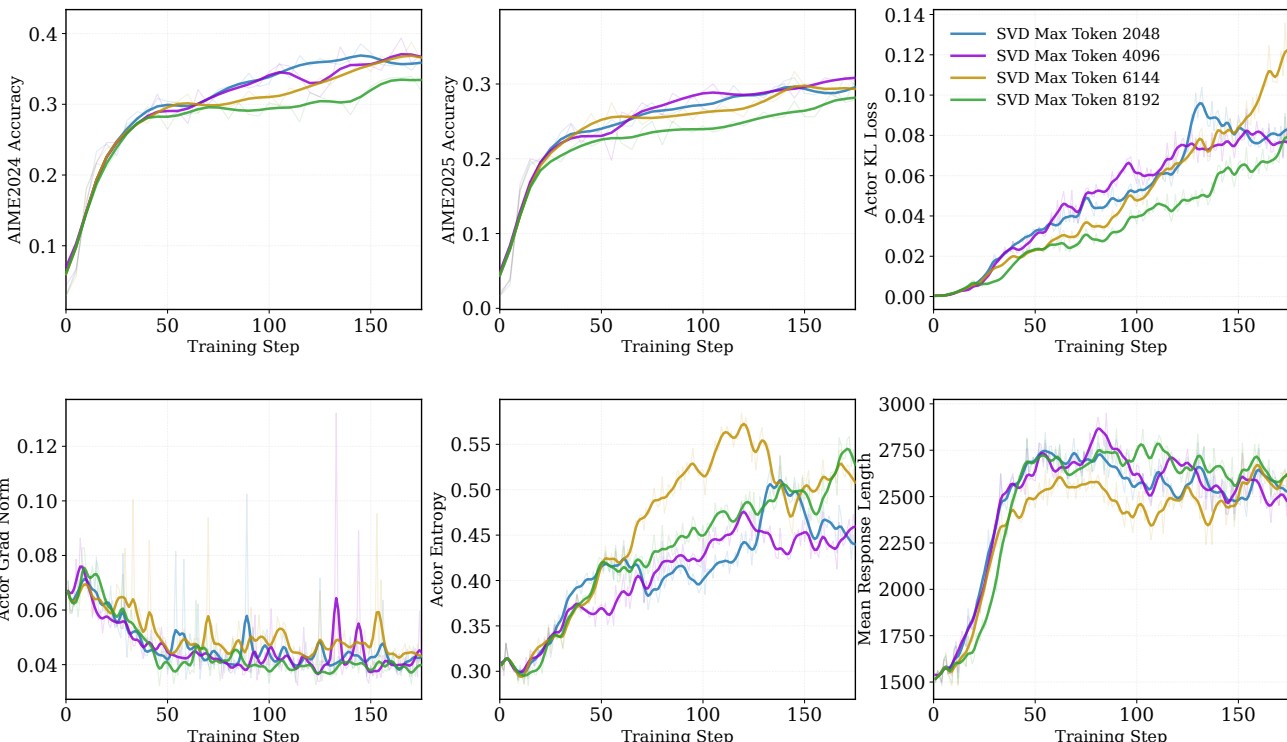

*Figure 10.* **Effect of the SVD subspace budget** $M_{\max}$ **on learning dynamics.** Ablations over $M_{\max} \in \{2048, 4096, 6144, 8192\}$ (rank $k$ fixed) under group rollouts ($G=4$) and max response length 4096. We report AIME2024/2025 accuracy (top row, left/middle), policy drift measured by actor KL loss (top row, right), optimization stability via actor gradient norm (bottom row, left), exploration via actor entropy (bottom row, middle), and mean response length (bottom row, right). Moderate budgets ($M_{\max}=2048$–4096) yield similar accuracy and stable optimization, consistent with the redundancy and low effective dimensionality of Transformer representations, which makes the dominant rank-$k$ positive subspace recoverable from a subsample. Increasing $M_{\max}$ beyond this range exhibits diminishing returns and can alter the gating signal: very large budgets may compress residual contrast and push token-wise weights toward their floor, weakening error-specific shaping (e.g., $M_{\max}=8192$ shows lower KL yet slower accuracy gains), while intermediate-large budgets can increase drift and exploration (higher KL/entropy) without a proportional accuracy improvement (e.g., $M_{\max}=6144$). Overall, $M_{\max}=4096$ provides a favorable tradeoff between subspace completeness, residual discriminability, and SVD compute.

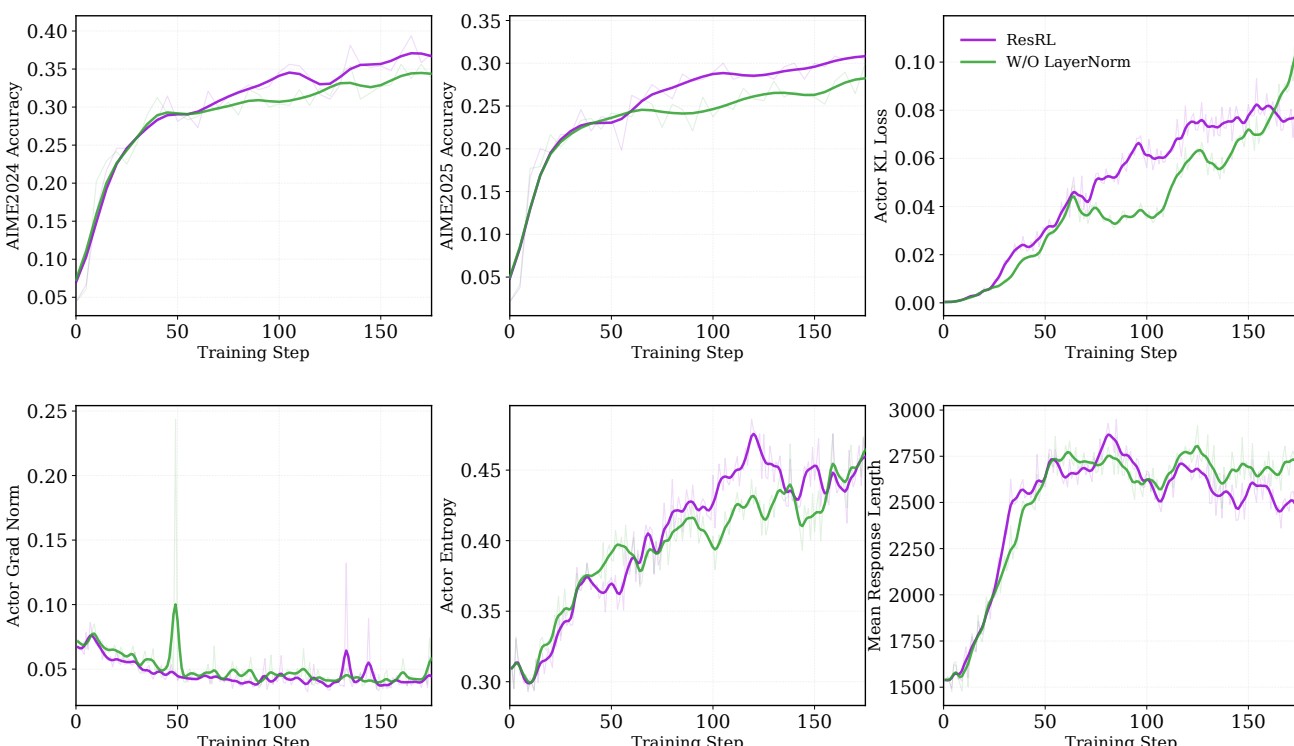

*Figure 11.* Impact of representation normalization on performance and stability. Removing the LayerNorm and centering mechanism (green) results in degraded performance on **(a)** AIME2024 and **(b)** AIME2025 compared to the standard ResRL configuration (purple). **(c)** The optimization dynamics reveal that the unnormalized configuration suffers from severe instability, evidenced by high-variance spikes in actor gradient norms. Crucially, this confirms that normalizing the representation geometry is essential for deriving reliable residual signals, thereby preventing erratic policy updates and ensuring robust optimization.

## D.2. Additional Analysis

**Pass@$k$ dynamics and compute regimes.** Figures 2, 3, and 5 visualize Pass@$k$ on AIME24/25 and AMC23 as the sampling budget increases (from low-$k$ to high-$k$). Across model scales, ResRL exhibits its most consistent advantage in the low-to-mid compute regime ($k \leq 64$), i.e., it improves Pass@1-style reliability and the early portion of the Pass@$k$ curve where practical decoding budgets typically operate. This pattern is aligned with the intended role of residual-based negative reweighting: by attenuating updates on negatives that are geometrically aligned with the positive subspace, ResRL avoids suppressing "innocent" intermediate steps that are shared across successful and failed rollouts, thereby improving sample efficiency under constrained sampling. Importantly, ResRL remains competitive at high $k$ (and can dominate in certain backbones), indicating that concentrating suppression in $S^{\perp}$ does not collapse exploration; rather, it reallocates negative pressure toward error-specific components while preserving diversity through the protected subspace.

**Mathematical reasoning (Avg@16).** Table 1 reports Avg@16 across six benchmarks on Qwen3-1.7B/4B/8B. ResRL improves the aggregate Avg@16 at all scales, with the largest and most diagnostic gains on AIME24/25. On Qwen3-1.7B, ResRL reaches an average of 48.6, exceeding NSR (47.5) and substantially improving over FlowRL (42.0) and GRPO (35.9); the improvement is concentrated on AIME24/25, while near-saturated datasets (e.g., MATH500) change minimally. On Qwen3-4B, ResRL achieves the strongest overall average (57.0), surpassing FlowRL (53.6), GRPO (53.1), and NSR (52.1); notably, it raises AIME24/25 to 45.2/38.6 (vs. 35.4/30.2 for FlowRL), consistent with improved reliability under fixed sampling (Avg@16) where indiscriminate negative upweighting can over-penalize partially-correct shared structure. On Qwen3-8B, ResRL attains the best average (64.7), improving AIME25 and MATH500 to 41.1 and 92.7 while remaining competitive on AMC23 and Olympiad. Interestingly, NSR can be higher on AIME24 at this scale, suggesting that diversity-centric penalties may still expand high-variance search on a subset of problems, but ResRL yields the strongest overall profile—consistent with the goal of reducing destructive suppression of shared reasoning directions while still penalizing genuinely erroneous components.

**Code reasoning.** Table 2 evaluates Qwen3-4B on LiveCodeBench, CodeForces, and HumanEval+. ResRL is best on all reported metrics: it improves LiveCodeBench to 43.2 Avg@16 and 59.9 Pass@16 (vs. 42.4/58.7 for FlowRL), and yields a clear margin on CodeForces with the highest rating and percentile (1469.5, 78.9), outperforming NSR (1340.9, 69.3) and FlowRL (1333.7, 68.7). HumanEval+ is near saturation for all strong methods, where ResRL reaches 97.0 Pass@16, matching or slightly exceeding the best baseline. Overall, the largest transfer signal is on the open-ended, distribution-shifted setting (CodeForces), consistent with the hypothesis that suppressing overlap-induced interference improves robustness beyond in-distribution Pass@$k$ gains.

**Long-horizon agent tasks.** Table 3 reports ALFWorld and WebShop results on Qwen2.5-7B-Instruct. On ALFWorld, ResRL improves overall success to 86.7, surpassing PPO (80.4), EMPG (78.5), and GRPO (74.8), with broad gains across sub-tasks (e.g., LOOK 85.5 and PICK2 84.2). This behavior supports the long-horizon intuition: successful and failed trajectories often share substantial prefixes, so naive negative reinforcement can corrupt reusable sub-policies; ResRL mitigates this by protecting shared directions and concentrating suppression on prefix-divergent components. On WebShop, ResRL improves success to 71.5 (vs. 69.3 for EMPG and 65.6 for GRPO) while maintaining a competitive task score (81.2), indicating improved completion reliability without sacrificing reward-bearing behaviors that require exploration.

**Tool-use robustness on BFCL.** Table 4 evaluates BFCL tool-use with multi-turn and single-turn accuracies. ResRL achieves the best Multi-Turn OA (41.25) and the highest overall accuracy (68.95). The gains are particularly pronounced on error-sensitive subsets: Miss Func improves to 47.0 and Miss Param to 34.0, consistent with reduced compounding of localized decision errors in tool selection and argument specification. At the same time, Long-Context remains substantially harder (e.g., 35.5 for ResRL), suggesting that the observed improvements arise primarily from mitigating localized decision errors and stabilizing multi-step tool planning, rather than extending context capacity per se.

**Design takeaway.** Across math, code, agents, and function calling tasks, the empirical profile is consistent with interference control in representation space: (i) the largest gains appear in regimes where shared partial structure between positives and negatives is prevalent (AIME and long-horizon trajectories), and (ii) improvements concentrate in reliability-centric metrics (Avg@16 and low-$k$ Pass@$k$), while remaining competitive at high $k$. These trends support the view that residual-projection reweighting reduces destructive negative-positive overlap without forcing premature mode collapse, providing a principled precision–diversity trade-off that is favorable under realistic compute budgets.

# E. Training Parameters

*Table 6.* Comprehensive Hyperparameter Configuration for ResRL Math Training. The table details model, optimization, generation, and infrastructure settings derived from the training script.

| Parameter | Value | Parameter | Value |
|---|---|---|---|
| *Model & Data Configuration* | | *Generation & Rollout (Inference)* | |
| Base Model | Qwen3-1.7B/4B/8B | Rollout Number ($N$) | 4 |
| Algorithm Estimator | GRPO | Temperature | 0.6 |
| Total Epochs | 1 | Top-p | 1.0 |
| Global Train Batch Size | 256 | Top-k | -1 (Disabled) |
| Max Prompt Length | 2048 | Thinking Template | False |
| Max Response Length | 4096 | Truncation Direction | Left |
| Truncation Mode | Left | Devices | Nvidia A100 |
| *Optimization Details* | | *SVD-based Exploration (Critical)* | |
| Learning Rate | $1 \times 10^{-6}$ | **SVD Rank** | **64** |
| LR Warmup Steps | 10 | **SVD Token Weighting** | **True** |
| Weight Decay | 0.1 | SVD Max Pos Tokens | 4096 |
| PPO Mini-Batch Size | 64 | | |
| KL Loss Coefficient | 0.0 | *Infrastructure & Parallelism* | |
| Entropy Coefficient | 0.0 | Tensor Parallel Size (TP) | 8 |
| Gradient Checkpointing | True | GPUs per Node | 8 |
| Dynamic Batch Size | False | GPU Memory Utilization | 0.65 |
| Remove Padding | True | Save Frequency | 50 Steps |

*Table 7.* Hyperparameter Configuration for ResRL Code Training. The table details model, optimization, generation, and infrastructure settings derived from the training script.

| Parameter | Value | Parameter | Value |
|---|---|---|---|
| ***Model & Data Configuration*** | | ***Generation & Rollout (Inference)*** | |
| Base Model | Qwen3-4B | Rollout Number ($N$) | 4 |
| Algorithm Estimator | GRPO | Temperature | 0.6 |
| Total Epochs | 1 | Top-p | 1.0 |
| Global Train Batch Size | 64 | Top-k | -1 (Disabled) |
| Max Prompt Length | 2048 | Thinking Template | False |
| Max Response Length | 8192 | Truncation Direction | Left |
| Truncation Mode | Left | Devices | Nvidia A100 |
| ***Optimization Details*** | | ***SVD-based Exploration (Critical)*** | |
| Learning Rate | $1 \times 10^{-6}$ | **SVD Rank** | **64** |
| LR Warmup Steps | 10 | **SVD Token Weighting** | **True** |
| Weight Decay | 0.1 | SVD Max Pos Tokens | 4096 |
| PPO Mini-Batch Size | 32 | | |
| KL Loss Coefficient | 0.0 | ***Infrastructure & Parallelism*** | |
| Entropy Coefficient | 0.0 | Tensor Parallel Size (TP) | 8 |
| Gradient Checkpointing | True | GPUs per Node | 8 |
| Dynamic Batch Size | False | GPU Memory Utilization | 0.65 |
| Remove Padding | True | Save Frequency | 50 Steps |

