# OpenReview forum: "ResRL: Boosting LLM Reasoning via Negative Sample Projection Residual Reinforcement Learning"
_ICML.cc/2026/Conference — ICML 2026 regular_

### Official Review · Reviewer_a534 · 2026-02-24

**Soundness:** 3
**Presentation:** 2
**Significance:** 2
**Originality:** 3
**Overall Recommendation:** 4
**Confidence:** 3

**Summary:**

This article strives to examine a central concept regarding the "gradient conflict" in RLVR. Specifically, it addresses the issue where algorithms like NSR indiscriminately penalize negative samples, thereby suppressing valid semantic structures shared with correct responses. Overall, this paper's fundamental contribution consists of "ResRL," a method that constructs a positive subspace from correct response hidden states and uses the projection residual of negative tokens to dynamically modulate penalties. This ensures that errors are penalized heavily while shared valid semantics are preserved.

**Compliance With Llm Reviewing Policy:**

Affirmed.

**Final Justification:**

ResRL introduces a novel method to mitigate semantic interference from negative samples in RLVR, supported by solid theoretical motivation and strong empirical results across benchmarks like AIME, LiveCodeBench, and ALFWorld. The rebuttal addresses my main concerns by providing runtime profiling, training dynamics, and hidden-layer ablations, showing that performance gains are not solely due to entropy stabilization but stem from structured suppression of destructive updates. Remaining limitations include computational cost from SVD, partial alignment between theory and implementation, and limited evaluation scope, which the authors should clarify in the final version. The rebuttal reinforces its positive contributions, so I recommend weak accept.

**Key Questions For Authors:**

1.It seems that the curve for ResRL on the training set is not provided. Do the training set metrics, such as entropy and length statistics, differ from the GRPO and NSR methods?

2.The paper aims to enhance model performance (Pass@1) while preserving output diversity (Pass@k). Methods that maintain or stabilize entropy to ensure generation diversity could potentially achieve similar outcomes. Could you provide a comparison with such methods and discuss the final results in terms of performance and diversity?

3.Matrix computations and SVD decompositions, which are time-consuming operations, are essential to ResRL. Do you have performance improvement curves that show the effect of ResRL with respect to time, rather than just the number of steps? Comparing results with the same number of steps might not be a fair comparison due to the computational overhead of these operations.

**Limitations:**

1.The paper only provides results in the "non-thinking" mode, without presenting corresponding results for the "thinking" mode, limiting the understanding of ResRL's performance across different operational modes.

**Strengths And Weaknesses:**

**Strengths**

1.This article introduces a method to reduce the interference from negative samples through residual projection, addressing the semantic overlap issue between positive and negative responses in traditional methods.

2.The authors link LLD to gradient interference and derive a theoretical upper bound using orthogonal-complement energy, providing solid justification for the method.

3.Experimental results demonstrate that ResRL outperforms baselines on various benchmarks, especially in tasks like AIME, LiveCodeBench, and ALFWorld, showing significant performance gains and stability in long-horizon training.

**Weaknesses**

1.Performing SVD for every prompt group introduces additional computational overhead compared to simple scalar operations in GRPO. The paper lacks a detailed wall-clock time comparison to quantify this cost.

2.The theoretical framework assumes linear gradient interference, but the practical implementation uses the penultimate hidden layer for feature representation, which may not fully align with this assumption.

**Typos**

L18: "...it significantly reduce" -> "...it significantly **reduces**"

L44: "...it inadvertently decrease" -> "...it inadvertently **decreases**"

L89: "...we employ low-rank approximation to construct representation space" ->  "...we employ low-rank approximation to construct **the** representation space"

L284 & L287: It should be "81.4" in bold instead of "81.2"

---

> ### Author Rebuttal · Authors · 2026-03-31
>
> We thank the reviewer for the detailed, practically oriented review and address each point below.
>
> > W1: SVD overhead — lack of wall-clock comparison
>
> We agree that asymptotic analysis alone is insufficient. We therefore report direct runtime profiling on the Qwen3-4B math setting on **8 NVIDIA H20 GPUs** (30-step average, identical hardware/batch/length):
>
> | Metric                    | FlowRL   | GRPO     | ResRL    |
> | ------------------------- | -------- | -------- | -------- |
> | Wall-clock time (s)       | 12142.56 | 12210.93 | 12326.80 |
> | Time per step (s)         | 404.60   | 406.88   | 410.75   |
> | Peak reserved memory (GB) | 145.45   | 146.13   | 143.37   |
> | Peak device memory (GB)   | 86.38    | 87.21    | 84.03    |
>
>
> On the profiled Qwen3-4B math setting, ResRL adds only **0.95% wall-clock time** relative to GRPO, with comparable **peak device memory** and no measurable increase in this profile.
>
> > W2: Theory assumes linear gradient interference but uses penultimate layer
>
> Lemma 1 is written for the representation $x$ immediately before the output head, whereas the algorithm uses the penultimate hidden layer as a motivated semantic proxy. This choice is empirical rather than theorem-exact: the final layer is more output-bound, the penultimate layer is semantically more stable, and the ablations show that it is the strongest overall late-layer default. We will make this distinction explicit in the revision.
>
> To further validate this choice, we have completed an extended **three-layer ablation on Qwen3-4B** (**4096-token budget, $k=64$**):
>
>
> | Hidden Layer (Avg@16) | AIME24   | AIME25   | AMC23    | Avg      |
> | --------------------- | -------- | -------- | -------- | -------- |
> | Final                 | 46.7     | 36.5     | 83.6     | 55.6     |
> | Third-to-last         | 44.2     | 39.5     | 85.1     | 56.3     |
> | **Penultimate**       | **45.2** | **38.6** | **89.4** | **57.7** |
>
>
> This suggests a viable late-layer neighborhood rather than a brittle exact layer choice, with the penultimate layer as the strongest default overall.
>
>
> > KQ1: Training set metrics — entropy, length statistics, ResRL curve
>
> Following the reviewer's suggestion, we provide a **direct GRPO / NSR / ResRL comparison** of training dynamics in [Rebuttal Figure 1](https://anonymous.4open.science/r/resrl_png-58B2).
>
>
> The figure reports actor entropy, log-scale gradient norm, response length, and AIME24/AIME25 score. ResRL stays much closer to GRPO than to NSR in entropy (about **0.34-0.41** vs NSR rising to **0.55+**), while avoiding the large gradient spikes seen in GRPO and keeping response length in a similar **2100-2500** token range.
>
> > KQ2: Comparison with entropy-stabilizing methods
>
> We agree that entropy-preserving/stabilizing methods are an important comparison family for the Pass@1–Pass@k trade-off. Recent studies such as GTPO (Simoni et al., 2025) and EMPG show that entropy control can improve exploration stability and mitigate policy collapse. GTPO also suggests that entropy control alone is insufficient, and that additional mechanisms are needed to address token-level conflicts. This is closely related to our motivation: in RLVR, the challenge is not only reduced entropy, but also destructive interference between negative and positive updates on semantically shared tokens.
>
> ResRL is complementary to this line of work. Instead of directly regularizing entropy, ResRL suppresses negative updates only in error-specific directions while preserving gradients aligned with the positive subspace. Notably, even with KL and entropy coefficients both set to zero in our settings, ResRL still maintains stable exploration and yields a better performance–diversity trade-off. On Qwen3-4B math, ResRL achieves 57.0 Avg@16, outperforming FlowRL (53.6), GRPO (53.1), and NSR (52.1), and improves Pass@128 by 7.0% over NSR averaged on AIME24/AIME25/AMC23. On long-horizon agents, it also surpasses the entropy-aware EMPG baseline on both ALFWorld (86.7 vs. 78.5) and WebShop (71.5 vs. 69.3).
>
> We therefore view entropy stabilization as complementary to ResRL rather than an alternative explanation of our gains.
>
> > KQ3: Performance improvement curves with respect to time
>
> Because the per-step gap is only **3.87s** (**406.88s/step** for **GRPO** vs **410.75s/step** for **ResRL**), **250** training steps correspond to roughly **28.26h** for GRPO and **28.52h** for ResRL, i.e., a difference of only about **16 minutes**. We therefore expect the time-aligned curves to preserve the same qualitative ordering, and will include the explicit time-axis plot in the revision.
>
> > Typos
>
> We are grateful to the reviewer for identifying these typographical errors. We have corrected all of them in the revised manuscript.
>
> ---
>
> We hope the added evidence clarify the remaining practical concerns. We will also note the current scope more explicitly in the revision: the math results here are limited to the no-think mode due to compute budget.

---

> > ### Author Rebuttal · Reviewer_a534 · 2026-04-01
> >
> > Thank you for the detailed response and clarifications. My concerns have been addressed, and I will maintain my positive score of 4.

---

> > > ### Author Response · Authors · 2026-04-02
> > >
> > > We are sincerely grateful for the reviewer’s careful follow-up and for taking the time to read the rebuttal in detail. We especially appreciate the acknowledgment that the rebuttal fully addressed the original concerns. Thank you again for the constructive and practically oriented feedback throughout the review process; it helped us substantially strengthen both the presentation and the empirical clarification of the paper.
> > >
> > > For clarity, the rebuttal resolved the main issues raised in the initial review. We added direct wall-clock and memory profiling on the Qwen3-4B math setting, showing that ResRL incurs only a small extra runtime cost on the profiled setup; we clarified the theory-to-implementation distinction and supported the practical layer choice with a three-layer ablation, which suggests a viable late-layer neighborhood rather than a brittle exact-layer dependency; we added direct GRPO/NSR/ResRL training-dynamics comparisons; and we quantified the wall-clock gap per step, making clear that, on this setting, the step-aligned and time-aligned comparisons are very close in practice.
> > >
> > > These clarifications build on the paper’s core strengths: the method addresses a meaningful weakness of negative-sample RLVR, provides a concrete residual-projection mechanism motivated by gradient-interference analysis, and shows gains across multiple settings including mathematical reasoning, code, and long-horizon agents.
> > >
> > > Given the reviewer’s indication that the original concerns have been fully resolved, we respectfully hope that the final assessment can reflect the strengthened post-rebuttal picture of the paper: a technically sound and empirically supported contribution, now accompanied by clearer practical evidence and a more complete explanation of why the method works.

---

### Official Review · Reviewer_hhhM · 2026-03-12

**Soundness:** 3
**Presentation:** 3
**Significance:** 3
**Originality:** 3
**Overall Recommendation:** 4
**Confidence:** 4

**Summary:**

This paper proposes ResRL, an RLVR method that builds a low-rank positive subspace from hidden states of successful tokens, then reweights negative-token updates based on how far they lie outside that subspace. The goal is to avoid uniformly penalizing negative responses when they still contain useful reasoning components shared with correct ones. Empirically, the authors report gains over GRPO/NSR across reasoning, coding, agent, and tool-use benchmarks

**Compliance With Llm Reviewing Policy:**

Affirmed.

**Final Justification:**

The paper presents a clear and original method for improving RLVR by selectively reweighting negative updates based on projection residuals from a positive hidden-state subspace. Its empirical evaluation is broad and strengthens the significance of the work. My main concerns were about robustness, the fairness of the math setting, and practical efficiency, and the rebuttal addressed them well through additional ablations, truncation analysis, larger-budget checks, runtime profiling, and clarification of the DAPO/GRPO comparison. This rebuttal improved my confidence in the soundness and practical value of the method and led me to raise my evaluation.

**Key Questions For Authors:**

No questions.

**Limitations:**

Yes

**Strengths And Weaknesses:**

**Strengths**

1. The paper is motivated by an interesting and well-defined problem. It identifies a plausible limitation of RLVR methods, namely that uniformly penalizing negative responses can also suppress semantic components shared with correct reasoning paths. The proposed ResRL design is well aligned with this motivation: it builds a low-rank positive subspace from token-level hidden states and uses projection residuals to selectively downweight error-specific negative updates.

2. The empirical evaluation is fairly comprehensive. The paper tests ResRL on multiple backbone scales and across a broad set of tasks, including mathematics, coding, agent benchmarks, and function calling. This breadth of evaluation makes the empirical claims more convincing and suggests that the method is not limited to a single benchmark or model setting.

**Weaknesses**

1. The method appears sensitive to several design choices, which may limit robustness. The ablations show that rank k has a clear protection–discrimination tradeoff: too small under-covers shared semantics, while too large leads to collapsed residual contrast and bursty gradients. Similarly, the choice of hidden layer, the SVD budget, and the normalization scheme all materially affect performance and stability. This suggests that ResRL may require nontrivial tuning rather than being a drop-in replacement for GRPO/NSR.

2. The experimental setting for mathematical reasoning needs further justification. The paper states that math training uses the DAPO training set with a 4096-token budget. And the reported results show that DAPO is substantially weaker than GRPO across the Qwen3-4B and Qwen3-8B math settings. Given that long reasoning traces are common in this regime, the paper should provide statistics on truncation, such as the fraction of rollouts that hit the response limit and how this varies by model scale. Without such evidence, it is difficult to assess whether the reported gains partly reflect an unfavorable setting for some baselines, especially DAPO.

3. The efficiency discussion remains incomplete from an empirical perspective. The paper does acknowledge the additional cost of ResRL and provides a theoretical complexity analysis for the extra per-group computation induced by subspace construction, truncated SVD, and residual evaluation. However, the paper does not report direct runtime or resource comparisons against GRPO/NSR, such as training time, throughput, or peak memory. Since ResRL introduces extra hidden-state processing for every rollout group, such empirical efficiency evidence would be important for understanding the practical cost-benefit tradeoff.

---

> ### Author Rebuttal · Authors · 2026-03-31
>
> We thank the reviewer for the detailed concerns and address each point below.
>
> > W1: Sensitivity to design choices may limit robustness
>
> In the current setting, the main ablated design choices are **k**, **q**, and the late-layer selection; Mmax and normalization behave as **stable defaults** in our tested regime. Our ablations suggest that these choices are interpretable and admit stable defaults across the tested settings: **$k=64$**, **$q=0.1$**, and the **penultimate layer**. Keeping these defaults already captures the main empirical gains in the tested settings and is sufficient to obtain strong improvements over the main baselines.
>
> - **Subspace construction**: Figure 4 shows that $k=64$ is the most accurate and stable setting across AIME24/25, indicating a broad stable region rather than a brittle optimum. Consistently, the rank-1 MeanRes-1D surrogate is **7.1** points worse on average, suggesting that the full SVD-based subspace is the more reliable default in our current pipeline (see [Rebuttal Figure 4/6](https://anonymous.4open.science/r/resrl_png-58B2)). Figure 10 further shows that $M_{\max}\in2048,4096$ yields comparable results, while Figure 11 shows that LayerNorm acts as a fixed prerequisite for comparable residual scales rather than a fragile tuning knob.
> - **Hidden layer**: We have now completed a full three-layer comparison on Qwen3-4B:
>
>   | Hidden Layer (Avg@16) | AIME24   | AIME25   | AMC23    | Avg      |
>   | --------------------- | -------- | -------- | -------- | -------- |
>   | Final                 | 46.7     | 36.5     | 83.6     | 55.6     |
>   | Third-to-last         | 44.2     | 39.5     | 85.1     | 56.3     |
>   | **Penultimate**       | **45.2** | **38.6** | **89.4** | **57.7** |
>
>   These results support a late-layer tradeoff and suggest a viable late-layer neighborhood rather than a brittle exact layer choice.
>
> > W2: Math setting — DAPO with 4096-token budget may unfairly handicap baselines
>
> In the math setting, we use a shared 4096-token response budget for computational tractability, with all reported runs conducted in **no-think mode** under this setup. Using the same budget controls one confound, but it does not by itself rule out differential sensitivity to truncation, so we add both cross-scale truncation statistics and an 8192-token robustness check. Across Qwen3-1.7B / 4B / 8B, the early-training response truncation ratio—defined as the fraction of rollouts that hit the 4096-token cap—is only **2.93% / 3.13% / 3.51%**, and the converged average response length is approximately **2500 / 2550 / 2450** tokens, respectively. This suggests that truncation is limited under the shared 4096-token setup. The full response-length and truncation ratio trajectories are shown in [Rebuttal Figure 3](https://anonymous.4open.science/r/resrl_png-58B2).
>
> Additionally, we provide a robustness check at 8192 tokens. To test whether the conclusion depends critically on the 4096 cap, we trained both DAPO and ResRL on **Qwen3-4B with an 8192-token response budget** and evaluated them under Avg@16 on the same benchmarks:
>
>
> | Method | AIME24 | AIME25 | AMC23 |
> | ------ | ------ | ------ | ----- |
> | DAPO   | 41.8   | 35.8   | 80.5  |
> | ResRL  | 48.8   | 44.0   | 87.7  |
>
>
> Under this larger-budget check, ResRL is higher than DAPO on all three benchmarks (**+7.0** on **AIME24**, **+8.2** on **AIME25**, and **+7.2** on **AMC23**). We treat this as a supplementary robustness check rather than a separate main result: it suggests that the relative ranking is not obviously reversed when the response budget is doubled, while the primary fairness argument still comes from the same-budget 4096-token comparison in Table 1.
>
> We will add these 8192-token results and the truncation statistics to the revised manuscript.
>
> > W3: Efficiency discussion lacks empirical evidence
>
> We have now completed direct runtime profiling on the Qwen3-4B math setting on **8 NVIDIA H20 GPUs**. Relative to GRPO, ResRL adds **0.95% wall-clock time** (115.87 seconds over 30 steps, about 3.86 seconds per step), with comparable **peak device memory** and no measurable increase in this profile. On this profiled setting, the added cost is therefore small.
>
>
> | Metric                    | FlowRL   | GRPO     | ResRL    |
> | ------------------------- | -------- | -------- | -------- |
> | Wall-clock time (s)       | 12142.56 | 12210.93 | 12326.80 |
> | Time per step (s)         | 404.60   | 406.88   | 410.75   |
> | Peak reserved memory (GB) | 145.45   | 146.13   | 143.37   |
> | Peak device memory (GB)   | 86.38    | 87.21    | 84.03    |
>
>
> The extra cost remains limited because the SVD is computed on a capped sampled subset with low target rank ($k=64$).
>
> ---
>
> We hope these additions help address the reviewer's concerns. Taken together, they suggest that ResRL admits stable defaults, that the observed gains are not well explained by truncation alone, and that the measured runtime overhead is small on the profiled setting.

---

> > ### Author Rebuttal · Reviewer_hhhM · 2026-03-31
> >
> > Thank you for the detailed rebuttal. My concerns have been addressed.

---

> > > ### Author Response · Authors · 2026-04-02
> > >
> > > We thank the reviewer for the thoughtful follow-up and the opportunity to clarify this point. Our claim is not that GRPO is generally stronger than DAPO; rather, the literature does not support a universal ordering, and the GRPO–DAPO ranking appears **configuration-sensitive**.
> > >
> > > Several recent studies already report settings in which GRPO is better than DAPO on the overall average, or at least on part of the evaluation suite. **MASA** reports on Qwen3-8B that GRPO exceeds DAPO on the six-benchmark average in both **Pass@1 (51.04 vs. 45.98)** and **Pass@32 (75.60 vs. 74.56)** [1]. **Plan Then Action Enhanced Reasoning** likewise finds GRPO higher than DAPO on MATH500, AIME24, AIME25, AMC23, and the overall average (76.85 vs. 74.77) on Qwen3-8B [2]. **Prompt Augmentation** reports a mixed pattern: GRPO is higher on AIME24 (25.3 vs. 20.7), while DAPO is only slightly higher on the benchmark-level average (**44.4 vs. 43.9**) [3]. **MathForge / Harder Is Better** is similar: GRPO is higher on AMC23 and OlympiadBench [4]. **MicroCoder** reports a Qwen3-4B, train-4K/test-8K setting where GRPO and DAPO are tied on the overall average (36.3 vs. 36.3), with each method better on different subsets [5]. Finally, **$\lambda$-GRPO** finds vanilla GRPO slightly higher than a DAPO-like token-mean variant on the Qwen2.5-3B average (**42.8 vs. 42.6**) [6]. We cite these papers only to show that the GRPO–DAPO ordering varies across settings.
> > >
> > > This is also consistent with the DAPO literature: its strongest gains are reported in native **long-CoT** settings with long horizons, while follow-up studies note **recipe sensitivity** across tasks and budgets [3,7,8].
> > >
> > > In our paper, the Qwen3-4B math comparison is a **controlled same-budget comparison**: the same training data, no-think setting, optimization budget, reward, rollout setup, and a shared **4096-token response cap**. Under this protocol, GRPO obtains a higher Avg@16 than DAPO. We therefore interpret the 4096 result as evidence about short-horizon behavior under matched conditions, rather than about each method’s best-case performance under its own native recipe. A plausible explanation is that GRPO’s group-relative normalization more directly favors relatively better sampled trajectories, which can help under a short response budget. By contrast, DAPO is explicitly designed to preserve and stabilize exploration in **long-CoT RL**, through ingredients such as **Clip-Higher**, **Dynamic Sampling**, **Token-Level Loss**, and **Overlong Reward Shaping** [8]. When the horizon is shortened, part of that exploration-preserving benefit may have less room to translate into completed correct solutions before truncation. We present this only as a regime-level interpretation, not as a definitive mechanism claim.
> > >
> > > Our new **8192-token** check on the same Qwen3-4B model reverses the ordering:
> > >
> > > | Method | AIME24 | AIME25 | AMC23 |
> > > |---|---:|---:|---:|
> > > | GRPO   | 33.1   | 31.3   | 74.8  |
> > > | DAPO   | 41.8   | 35.8   | 80.5  |
> > > | ResRL  | 48.8   | 44.0   | 87.7  |
> > >
> > > DAPO is higher than GRPO on all three benchmarks: **+8.7** on AIME24, **+4.6** on AIME25, and **+5.7** on AMC23, for a **+6.3-point average**. This does not prove the full mechanism, but it directly shows that the 4096 gap is **horizon-sensitive**, rather than evidence that the DAPO baseline was run incorrectly. This interpretation is also consistent with the fact that the 4096 comparison keeps the reward shared across baselines and does **not** enable DAPO-specific overlong shaping, so it isolates matched-budget behavior rather than each method’s best-case recipe.
> > >
> > > In revision, we will present the 4096 main table as a controlled short-budget comparison and the 8192 result as a robustness check of regime sensitivity. We hope this clarification and the new evidence address the reviewer’s concern and make the significance of our comparison clearer.
> > >
> > > [1] Kim et al., **Meta-Awareness Enhances Reasoning Models: Self-Alignment Reinforcement Learning**, arXiv 2025.
> > >
> > > [2] Dou et al., **Plan Then Action Enhanced Reasoning: High-Level Planning Guidance Reinforcement Learning for LLM Reasoning**, arXiv 2025.
> > >
> > > [3] Lu et al., **Prompt Augmentation Scales up GRPO Training on Mathematical Reasoning**, arXiv 2026.
> > >
> > > [4] Dai et al., **Harder Is Better: Boosting Mathematical Reasoning via Difficulty-Aware GRPO and Multi-Aspect Question Reformulation**, ICLR 2026.
> > >
> > > [5] Li et al., **Breaking Training Bottlenecks: Effective and Stable Reinforcement Learning for Coding Models**, arXiv 2026.
> > >
> > > [6] Wang et al., **$\lambda$-GRPO: Unifying the GRPO Frameworks with Learnable Token Preferences**, arXiv 2025.
> > >
> > > [7] Lian, **Comparative Analysis and Parametric Tuning of PPO, GRPO, and DAPO for LLM Reasoning Enhancement**, arXiv 2025.
> > >
> > > [8] Yu et al., **DAPO: An Open-Source LLM Reinforcement Learning System at Scale**, NeurIPS 2025.

---

### Official Review · Reviewer_hjsV · 2026-03-12

**Soundness:** 3
**Presentation:** 3
**Significance:** 3
**Originality:** 3
**Overall Recommendation:** 5
**Confidence:** 4

**Summary:**

ResRL addresses semantic overlap between positive and negative responses in RLVR training. It constructs an SVD-based low-rank subspace from positive-token hidden representations and computes the projection residual of each negative token. Tokens aligning closely with the positive subspace receive reduced penalty weights; tokens in the orthogonal complement receive stronger penalties. The paper provides a theoretical framework linking Lazy Likelihood Displacement (LLD) to gradient interference and shows orthogonal-complement energy serves as a proxy for representation alignment. Experiments on twelve benchmarks (math, code, agents, function calling) using Qwen models from 1.7B to 32B demonstrate improvements over GRPO, NSR, and other baselines.

**Compliance With Llm Reviewing Policy:**

Affirmed.

**Final Justification:**

The rebuttal addressed my main concerns with concrete additional evidence. I therefore maintain my positive evaluation and accept recommendation.

**Key Questions For Authors:**

1. What is the empirical distribution of $\frac{|\langle \delta^-, \delta^+ \rangle|}{|\langle x^-, x^+ \rangle|}$ across training? If the logit term varies by orders of magnitude, the proxy may not reliably rank tokens.

2. How often does the $|P|=0$ fallback trigger on harder benchmarks such as AIME?

3. Have you considered computing the positive subspace across multiple groups or using an EMA of $V_k$?

4. How does ResRL compare to simpler token-level reweighting methods without the SVD machinery?

**Limitations:**

The paper could discuss the logit-space factor gap more explicitly. Variance on small-sample benchmarks (e.g., AIME) and potential sensitivity to group composition also deserve acknowledgment.

**Strengths And Weaknesses:**

**Strengths:**
- Well-motivated problem: penalizing negative tokens can suppress semantically shared valid tokens. The LLD connection is concrete
- Clean theoretical decomposition. Lemma 1 (gradient inner-product factorization) connects gradient interference to representation alignment, motivating the use of orthogonal-complement energy as a proxy.
- Twelve benchmarks across four domains (math, code, agents, tool use), 1.7B to 32B. Pass@k curves show simultaneous gains in low-k accuracy and high-k diversity
- Thorough ablations: rank selection, hidden layer choice, quantile thresholds, SVD budget, LayerNorm, KL penalty, length-scaled rewards, each with training dynamics
- Computational overhead well-analyzed (Appendix C) with an $M_{max}$ budget cap to control SVD cost.

**Weaknesses:**
- The logit-space factor |<delta^-, delta^+>| is uncontrolled. If it dominates or varies substantially, the representation-based proxy could be misleading. No empirical decomposition of gradient interference into logit vs. representation terms
- Theoretical bound (Theorem 1) is loose and the algorithm does not directly optimize it. The connection from bound to weight formula (Eqs. 15-17) involves heuristic choices (quantile normalization, linear interpolation) justified by ablation, not theory
- Positive subspace assumption may not hold: positive trajectories can contain incorrect steps, negative trajectories can contain correct steps. No analysis of noisy positive labels on subspace quality
- Table 1 reports Avg@16 without confidence intervals. Only Table 3 mentions 3 seeds. AIME24/25 have 30 problems each; small accuracy differences could be noise
- No comparison against simpler token-level reweighting (e.g., log-probability difference or cosine-similarity weighting without SVD), making it difficult to isolate the contribution of the projection-residual mechanism.

---

> ### Author Rebuttal · Authors · 2026-03-31
>
> We thank the reviewer for the constructive review and address each point below.
>
> > W1 / KQ1: The logit-space factor $|\langle\delta^-,\delta^+\rangle|$ is uncontrolled
>
> ResRL does not control the full cross-sign interference term; it uses $e(x^-)$ only as a one-sided proxy for the representation-side factor in the exact head-gradient factorization. We therefore tested whether the logit-side term could overturn the representation-based ranking. In a **10-step Qwen3-4B math analysis run** (4160 sampled positive-negative token pairs), $\log_{10}\frac{|\langle \delta^-, \delta^+\rangle|}{|\langle x^-, x^+\rangle|}$ has step-level **mean / median / p90** of **−3.043 / −3.138 / −2.878**, with standard deviations **0.630 / 0.654 / 0.579**; the largest observed step-level p90 is **−1.956** (about $1.1 \times 10^{-2}$). This suggests the logit-space factor is much smaller than the representation factor, making reordering unlikely.
>
> > W2: Theorem 1 is loose; Eqs. (15)–(17) involve heuristic choices
>
> **Lemma 1** gives the exact factorization $\langle \nabla_W \ell^-, \nabla_W \ell^+ \rangle = \langle \delta^-, \delta^+ \rangle \cdot \langle x^-, x^+ \rangle$, showing that cross-sign interference includes a representation-space term. **Theorem 1** then uses the normalized orthogonal-complement energy $e(x^{-}) = \frac{1}{d}\lVert x^{-} - V_k V_k^\top x^{-} \rVert_2^2$ as a monotone one-sided proxy for overlap with the positive subspace $\mathrm{span}(V_k)$: under its assumptions, larger $e(x^-)$ implies smaller overlap, so an increasing mapping from $e(x^-)$ to the negative penalty remains theoretically conservative. **Eqs. (15)–(17)** instantiate this via quantile normalization and linear interpolation to $[w_{\min}, 1]$, chosen for stability across heterogeneous prompt groups rather than uniquely implied by the theory. A monotone mapping preserves the one-sided ordering.
>
> > W3: Positive subspace assumption may not hold under noisy labels
>
> We do not assume every positive token is correct; rather, we use the dominant low-rank geometry of positive-token representations as a prompt-conditional reference. In practice, the current pipeline already mitigates label noise through: (1) **LayerNorm + centering** (Eq. 12), which stabilizes feature scale; (2) the **SVD budget** $M_{\max}$, with at most $M_{\max}=4096$ sampled positive tokens and stable results for $M_{\max}\in\{2048,4096\}$ in Fig. 10; (3) **quantile-based normalization** (Eqs. 15--17), which relies on within-group ranking rather than absolute residual magnitude; and (4) the **$|P|=0$ fallback** (Algorithm 1), where ResRL reverts to NSR instead of constructing a subspace from noise. This is also consistent with two ablations: **rank ablation** (Fig. 4), where $k=64$ is a stable default, and **SVD budget ablation** (Fig. 10), which shows stability across different $M_{\max}$ values.
>
> > W4: Confidence intervals on small benchmarks
>
> We recomputed **per-problem bootstrap CIs** and **paired bootstrap** on the same 4B checkpoints under **Avg@8**. ResRL consistently outperforms NSR: **AIME24** 46.25 [31.25, 62.08] vs. 38.75 [25.00, 53.33], **AIME25** 40.42 [26.25, 55.42] vs. 33.33 [19.17, 49.17], **AMC23** 90.00 [81.88, 96.25] vs. 78.44 [68.12, 87.50], and **Avg** 58.89 [51.74, 66.25] vs. 50.17 [42.71, 57.81]. Paired bootstrap also favors ResRL on all three benchmarks: **+7.5** on **AIME24**, **+7.1** on **AIME25**, and **+11.6** on **AMC23**, with **Pr(diff > 0) > 97.5%** in each case. This supports the same qualitative conclusion.
>
> > W5: No comparison against simpler reweighting
>
> To isolate the effect of the SVD-based positive subspace, we add a **MeanRes-1D baseline** that replaces the multi-directional low-rank subspace with a single rank-1 mean direction while keeping normalization, quantile mapping, and the objective unchanged. Under Avg@16, ResRL outperforms MeanRes-1D in overall average at both tested scales ([Rebuttal Figure 4/6](https://anonymous.4open.science/r/resrl_png-58B2)) : **38.4 vs. 31.3** (1.7B) and **57.7 vs. 50.6** (4B). We also test a **4B cosine-similarity weighting without SVD**, which reaches **53.4**. These results suggest that the multi-directional SVD subspace is more effective than a single mean direction and a uniformly sampled cosine-similarity method.
>
> > KQ2: Frequency of the $|P|=0$ fallback
>
> As an early-training diagnostic, the average fallback ratio is 54.7% over the first 10 steps, and decreases as the average number of positive samples per group increases.
>
> > KQ3: Cross-group / EMA positive subspace
>
> Our design is prompt-local, since ResRL targets overlap between positive and negative trajectories within the same prompt group. A global or EMA-based subspace may reduce variance when positives are scarce, but could also blur prompt-specific directions. We will clarify this tradeoff in the revision.
>
> ---
>
> We appreciate the push for a tighter theory-to-algorithm account and will state the limitations more explicitly in the revision.

---

> > ### Author Rebuttal · Reviewer_hjsV · 2026-04-01
> >
> > The rebuttal addresses all five weaknesses with concrete evidence. The bootstrap CIs and paired tests (W4) confirm statistical significance on small benchmarks (Pr(diff > 0) > 97.5%). The MeanRes-1D and cosine-similarity baselines (W5) isolate the contribution of the multi-directional SVD subspace. The empirical characterization of the logit-space factor magnitude (W1) is informative. I maintain my original score.

---

> > > ### Author Response · Authors · 2026-04-04
> > >
> > > We sincerely thank the reviewer for this careful and thoughtful acknowledgment. We especially appreciate that you engaged with the rebuttal at a technical level and recognized the role of the new evidence in addressing the core concerns. Your summary accurately captures the main points: the rebuttal did not rely on general clarifications alone, but added targeted empirical support, including the logit-space factor analysis, statistical validation on small benchmarks, and simpler reweighting baselines that isolate the contribution of the multi-directional SVD subspace.
> > >
> > > We are particularly grateful that you viewed these additions as concrete and informative. We took the rebuttal very seriously and aimed to answer each weakness with direct evidence rather than broad claims. We believe this response also reflects the overall quality of the paper: the work is built on a well-motivated problem, a clean theoretical decomposition, broad empirical coverage, and careful ablations, and the rebuttal further strengthened the paper by making several previously implicit points explicit and quantitatively verified.
> > >
> > > Thank you again for the time, care, and technical attention you devoted to evaluating our submission. Your feedback has been genuinely valuable in helping us sharpen both the presentation and the empirical grounding of the paper.

---

### Official Review · Reviewer_qAK3 · 2026-03-13

**Soundness:** 4
**Presentation:** 3
**Significance:** 3
**Originality:** 2
**Overall Recommendation:** 4
**Confidence:** 4

**Summary:**

ResRL improves RL-based LLM reasoning by penalizing only the error-specific parts of negative samples. It builds a low-rank positive subspace from hidden states, uses projection residuals to reweight negative gradients, and consistently improves both accuracy and diversity across math, code, agent, and tool-use benchmarks.

**Compliance With Llm Reviewing Policy:**

Affirmed.

**Final Justification:**

This paper was already performing reasonably well in all aspects other than Soundness. The rebuttal addressed all my concerns. I have raised the score for Soundness from 2 to 4.

**Key Questions For Authors:**

- Given that all experiments in the paper were conducted using Qwen, does ResRL remain effective when applied to other model families?
- In practice, how much does the additional computational overhead introduced by ResRL increase the training time?
- How does ResRL perform if the output of a different layer—such as the third-to-last layer—is used as the input *x*?

**Limitations:**

Limitations are not discussed in this paper. Here are some suggestions for improvement:

The authors could primarily supplement the explanations and experiments related to the questions. In addition, two minor suggestions are offered: visualize the correlation between different parts of the negative samples and the positive samples; and explain why the improvement in ResRL performance at Qwen3 1.7B compared to the optimal baseline NSR is not significant.

**Strengths And Weaknesses:**

- In terms of Soundness, this work is generally excellent; for instance, the main experiments span a wide range of domains and include numerous highly reasonable ablation studies—such as the ablation regarding the choice of *k* for rank-*k* approximation. However, there are some weaknesses:
    - It lacks validation across a broader range of model families.
    - It lacks comparative experimental results regarding the training times of the different methods.
    - It analyzes only the outputs of the last and second-to-last layers (treated as *x*), lacking an analysis of additional layers.
- In terms of Presentation, this work is excellent and possesses many strengths; for example, Figure 1 is exceptionally well-executed—clear, intuitive, and easy to understand.
- Regarding Significance and Originality, this work is reasonably adequate, presenting no overly obvious weaknesses.

---

> ### Author Rebuttal · Authors · 2026-03-31
>
> We thank the reviewer for the supportive assessment and constructive questions, and address them below.
>
> > Q1: Does ResRL remain effective on non-Qwen model families?
>
> Our experiments already cover Qwen3 at 1.7B / 4B / 8B / 32B and Qwen2.5-7B-Instruct on agent tasks, showing transfer across scales and fine-tuning stages within the Qwen line.
>
> To directly address the concern beyond the original Qwen3 line, we additionally evaluated **MIMO-7B-Base** on the same **Avg@16** math protocol:
>
>
> | Method (Avg@16) | AIME24    | AIME25    | Avg       |
> | ----------------------- | --------- | --------- | --------- |
> | NSR                     | 30.42     | 28.13     | 29.28     |
> | GRPO                    | 31.25     | 26.04     | 28.65     |
> | **ResRL**               | **33.13** | **28.13** | **30.63** |
>
>
> On this additional 7B backbone, ResRL improves over GRPO by **+1.88 points** on **AIME24** and **+2.09 points** on **AIME25**, and it also exceeds NSR on **AIME24** while matching it on **AIME25**. The absolute level remains model-dependent, but the relative ranking remains favorable to ResRL, supporting transfer beyond the original Qwen3 math setup.
>
> This is also consistent with the method's premise: ResRL relies only on a linear output head and semantically meaningful late-layer representations, both standard in mainstream Transformer LLMs. The MIMO-7B-Base result provides additional empirical support for this view.
>
> > Q2: How much does the additional computational overhead increase training time?
>
> We profiled runtime on the Qwen3-4B math setting. Since **GRPO** and **NSR** share the same training path in our implementation and differ only in the scalar advantage coefficient, their runtime is effectively the same; we therefore report **GRPO** as representative. The table below reports 30-step averages on **8 NVIDIA H20 GPUs** under identical hardware, batch size, and sequence length:
>
>
> | Metric                       | FlowRL   | GRPO     | ResRL    |
> | ---------------------------- | -------- | -------- | -------- |
> | Wall-clock time (s) | 12142.56 | 12210.93 | 12326.80 |
> | Time per step (s)            | 404.60   | 406.88   | 410.75   |
> | Peak reserved memory (GB)    | 145.45   | 146.13   | 143.37   |
> | Peak device memory (GB)      | 86.38    | 87.21    | 84.03    |
>
>
> On the profiled Qwen3-4B math setting, ResRL adds only **0.95% wall-clock time** relative to GRPO, with comparable **peak device memory** and no measurable increase in this profile. We therefore characterize the extra cost as **small on this profiled setting**. The SVD-based subspace construction uses a small sampled subset ($M_{\max}=4096$ tokens) and a low target rank ($k=64$), so it contributes only a minor fraction of the overall forward-backward cost.
>
> > Q3: How does ResRL perform with a different layer (e.g., the third-to-last)?
>
> The paper already compares the penultimate and final hidden layers. We now add a **third-to-last-layer ablation on Qwen3-4B** under identical conditions (**4096-token budget, rank $k=64$**):
>
>
> | Hidden Layer (Avg@16) | AIME24   | AIME25   | AMC23    | Avg      |
> | --------------------- | -------- | -------- | -------- | -------- |
> | Final                 | 46.7     | 36.5     | 83.6     | 55.6     |
> | Third-to-last         | 44.2     | 39.5     | 85.1     | 56.3     |
> | **Penultimate**       | **45.2** | **38.6** | **89.4** | **57.7** |
>
>
> The pattern is not monotonic: the final layer is competitive on AIME24, the third-to-last layer is strongest on AIME25, and the penultimate layer is strongest on AMC23 and in overall average. This is consistent with a late-layer tradeoff between semantic abstraction and output-space coupling. Overall, several late layers are viable, with the penultimate layer as the strongest default overall. We will include this three-layer comparison in the revision to further strengthen our ablation analysis.
>
> ---
>
> In particular, we find that 1.7B remains more negative-dominated than 4B/8B ([Rebuttal Figure 2](https://anonymous.4open.science/r/resrl_png-58B2)), with fewer positive and more negative samples available per prompt group throughout training. By contrast, 4B and 8B move much earlier into a more positive-rich regime. Since ResRL constructs a prompt-local positive subspace from successful trajectories, the 1.7B setting provides both fewer anchors and a noisier estimate of the shared positive geometry, which naturally reduces the extra headroom over NSR. We therefore view the smaller 1.7B gain as a scale-related limitation of subspace estimation quality, rather than evidence against the mechanism itself.
>
> We hope these additional experiments and clarifications help resolve the remaining questions on generalization, efficiency, and layer sensitivity, and are useful for the final evaluation. We will also make the limitations discussion more explicit with analysis and visualization.

---

> > ### Author Rebuttal · Reviewer_qAK3 · 2026-04-02
> >
> > Many thanks to the author for the detailed response. All my questions regarding Soundness have been resolved. I will raise the Soundness score.

---

> > > ### Author Response · Authors · 2026-04-04
> > >
> > > We sincerely thank the reviewer for the positive acknowledgment and for confirming that our rebuttal has fully resolved the soundness concerns. We greatly appreciate your careful reading of the paper, your constructive questions regarding cross-family generalization, computational overhead, and hidden-layer choice, and your recognition of the additional evidence provided in the rebuttal. In the revision, we will incorporate the newly added non-Qwen baseline results, the runtime and memory profiling, and the extended hidden-layer ablation, and we will also clarify the analysis of the relatively smaller gain at 1.7B. We are grateful for your time and thoughtful evaluation, which have helped strengthen both the empirical support and the presentation of the paper.

---

### Decision · Program_Chairs · 2026-04-30

**Decision:**

Accept (regular)

**Comment:**

The reviewers indicate that the paper is a technically solid paper addressing a well-motivated problem, namely that penalizing negative tokens can suppress semantically shared valid tokens. The thorough ablations and Pass@k gains further strengthen the empirical evidence. While concerns remain regarding a loose theoretical bound, heuristic choices, computational overhead, and lack of runtime comparison, the reviewers generally agree these weaknesses are limited.
Based on the feedback from reviewers, the decision was made to recommend it for acceptance. We congratulate the authors on their acceptance!